# Building momentum: A computational account of persistence toward long-term goals

Sneha Aenugu [1]*, John P. O'Doherty[1]

Division of the Humanities and Social Sciences, California Institute of Technology, Pasadena, California, United States of America

* saenugu@caltech.edu

**Data availability statement:** Data and code used in the study is available at https://github. com/asneha213/goal-switching/tree/main.

## Abstract

Extended goals necessitate extended commitment. We address how humans select between multiple goals in a temporally extended setting. We probe whether humans engage in prospective valuation of goals by estimating which goals are likely to yield future success and choosing those, or whether they rely on a less optimal retrospective strategy, favoring goals with greater accumulated progress even if less likely to result in success. To address this, we introduce a novel task in which goals need to be persistently selected until a set target is reached to earn an overall reward. In a series of experiments, we show that human goal selection involves a mix of prospective and retrospective influences, with an undue bias in favor of retrospective valuation. We show that a goal valuation model utilizing the concept of 'momentum', where progress accrued toward a goal builds value and persists across trials, successfully explains human behavior better than alternative frameworks. Our findings thus suggest an important role for momentum in explaining the valuation process underpinning human goal selection.

## Author summary

Goals take time to accomplish and require commitment over extended periods of time. However, over time, some goals may become less attainable than others, necessitating switching between goals as contexts and circumstances change. A fundamental question concerns how humans switch goals across extended intervals. One possibility is that humans prospectively commit to specific goals in a way that is maximally sensitive to the likelihood of achieving that goal in the near future. Alternatively, humans might retrospectively persist in goals that they have previously worked toward. Consistent with this latter possibility, we found evidence that humans are retrospectively biased towards goals that they have spent time building progress in, even when it is more optimal to switch. We account for such a preference for accrued progress using a computational

**Funding:** This work was supported by a grant from the ARO MURI project W911NF2110328 to JO'D. SA and JO'D have received salary support from this funding. SA's salary is also supported by the Tianqiao & Chrissy Chen Center for social and decision neuroscience. The funders had no role in study design, data collection and analysis, decision to publish or preparation of the manuscript.

**Competing interests:** The authors have declared that no competing interests exist.

model incorporating the concept of momentum borrowed from classical mechanics. We show that momentum steadily integrates reinforcement throughout the goal progress and maintains stable goal commitment toward the goal despite environmental shifts. Although this results in suboptimal performance in our experimental paradigm, we show that momentum computations for goal commitment can have adaptive advantages for goal pursuit in real-world scenarios.

## Introduction

Humans often direct sustained effort to reach rewarding states that are distant from them. We often refer to such distant setpoints as goals – desired future states for our future selves. Whether it is climbing Mount Whitney or landing a dream job, these goals require sustained effort to ensure gradual progress toward the respective goals [1,2]. We often toggle between multiple such end states. For example, we wish to finish our project deadlines, while also contemplating unfinished construction projects at home. How do we select which long-term goal to pursue at any given point, and how do we assign value to multiple goals at the same time?

While a desired end state can be a source of motivation, maintaining such a long term goal could require intermediate reinforcement, signaling whether a target is achievable or not worth pursuing. Progress towards the target is a potential reinforcement signal that serves to validate our efforts toward goal pursuit. Sustained progress that draws us closer to the target motivates us to expend more effort in the pursuit, whereas a slump in progress lowers our drive and induces us to pursue another target that is more attainable. [3] found that even progress not explicitly signaled in a task can, when implicitly ruling out alternative options, influence autonomous goal selection by limiting exploration [4].

Here, we introduce a computational model of momentum in progress, to capture the notion of a sustained drive towards a distant goal. We propose that intermittent reinforcement signaling progress builds up in value and drives persistence towards goals. According to this framework, a goal that yielded sustained progress towards a target in the recent past can be perceived as more valuable than an alternative goal which made no progress in the past, but yet has a greater probability of reaching the target in the near future. Our proposed model of persistent valuation predicts an inertial effect in the willingness to change targets even when the rate of progress toward a goal comes to a standstill [5].

Momentum is a concept deeply entrenched in the domain of Newtonian physics that formalizes inertia or resistance to change that is fundamental to all physical quantities embodied with mass. In physics, momentum is defined as a product of mass and velocity, $p = m * v$, and within its formulation there is an evident corollary that the greater the momentum of an object, the greater is its resistance to change. In other words, more force is required to stop an object with higher momentum, which is a function of both its mass and velocity of progress.

Several applications of this concept in cognitive and computational science indicate that this construct might have implications beyond rigid body dynamics. In the perceptual decision-making literature, it has been observed that memory for the location of a moving target is often displaced in the direction of the target's motion, which has led to the use of the term *representational momentum* alongside the hypothesis that the perceptual system evolved to function in the regime of Newtonian physics [6–8]. In operant conditioning, the persistence of learned operant behavior in the face of altered conditions was termed *behavioral momentum* [9–13]. Momentum is also a key concept in machine learning utilized for stability in learning and optimization [14]. Momentum induces inertia in gradient updates

by maintaining a running average of recent updates instead of fluctuating momentary gradients to update expectations/policies. This rewards consistency in signed gradient errors, thereby maintaining stability in the direction of updates. The same concept was proposed as a possible explanation for the effect of mood on decision making [15,16].

Psychologists have previously linked the concept of momentum to the phenomenology of goal pursuit [17–19]. However, most of this previous work has been limited to qualitative formulations, introducing momentum as a potential account for the tendency of dispositional states to carry forward through time; task-set inertia, mental simulation, and rumination are a few examples. Moreover, momentum, as a term, has been a part of the popular vernacular relating to the pursuit of goals ("this project needs more momentum", "this fight is gathering momentum every day"). However, its quantitative formalization in the decision-making literature is lacking.

Decision inertia and persistence of actions are extensively studied in psychology, albeit in the context of intermittent reinforcements. Notable studies in the foraging literature have used the concept of the opportunity cost of time as a means to decide whether to continue foraging in the current patch or move to a new one with a delay in time [20–22]. Differently from this approach, in the present study we focus on scenarios where rewards are distant, which requires the maintenance of a persistent drive toward goal pursuit.

Suboptimal human persistence has been extensively documented in prior literature where there are sunk costs [23–25]. Humans are known to overpersist on goals that have incurred sunk costs of time, effort, and resources, even when it is optimal to abandon them. The sunk cost fallacy demonstrates similar persistence effects in relation to extended goals, the neural basis of which was recently investigated in humans [26]. But not all goal-selection decisions made in daily life occur in the presence of explicit sunk costs. For instance, we can decide to stop working on debugging code and rest instead, with the full knowledge that our work can be resumed later. In the present study, we explicitly emphasized investigating human goal persistence patterns in a multitasking setting rather than a sunk-cost one. Therefore, we allowed goals to retain accrued progress having switched away from them. This was done to mitigate the influence of sunk costs in our experiments related to abandoned goals completely losing their value. Consequently, the present study deviates from previous work [26], as sunk costs are likely to exert a potent influence on the long-term pursuit of the goal. Moreover, in [26], while overpersistence was observed in behavior in that study, providing a computational account for the overpersistence effect was not their focus. Here, our overall goal is to understand the nature of the overpersistence effect, particularly after the effects of sunk costs explicitly related to a loss of prior goal progress have been reduced.

We make two significant contributions through the study described here. Firstly, we introduce a novel paradigm to investigate the temporal nature of the valuation of extended goals in the presence of competing alternatives, a hitherto unexplored question. Secondly, we provide an algorithmic account for persistence in extended goals. We introduce momentum as a key computational variable driving inertia in goal valuation, providing a fresh perspective on human persistence and decision inertia.

## Results

### Suits task: A paradigm for studying extended goal pursuit

We introduce a suit collection paradigm to simulate extended goal pursuit inspired by traditional card games. A suit in the game comprises a specific number (say seven) of tokens of the same kind. The objective for the participants is to collect as many suits as they can and win points in the game (which ultimately translates to a monetary bonus). There are three

different types of suits available (*cat*, *hat*, *car*) and participants can freely switch between collecting different suits without penalty to earn points in the game. Participants only receive points when a suit is completed without any partial credit for any incomplete suits.

## Gameplay

The game is played in blocks with each block having 30 rounds. A sample round of the gameplay is shown in Fig 1. Participants start with an empty set of slots (7 for each token type as shown in Fig 1A).

**State:** The game state is indicated by the current slot configuration which indicates progress made toward different suits. An example slot configuration occurring a few rounds into the game is shown in Fig 1C.

**Actions:** There are 6 cards in the game, each corresponding to a specific suit type (two cards are allotted for each suit, see Fig 1B). On each round, a participant is shown three cards (one card for each suit). Participants can choose to flip between one of the three cards shown to receive tokens for a suit (see Fig 1C). For example, if a participant wishes to acquire a *car* token to collect the *car* suit, they would pick the *key* or the *baggage* card (whichever is available in the round) to try to acquire the token.

**Outcomes:** If the participant flips the *car*-related card (say), they receive the *car* token with a specific (non-zero) probability and the *hat* and the *cat* tokens with zero probability. If the participant receives the *car* token, one of the empty *car* slots is filled and the participant proceeds to the next round of token collection. If a suit's target is reached (say 7 *tokens* of one type are collected), the participant will receive 10 points for finishing the suit and all of those 7 slots are emptied for a fresh bout of collection.

**Game blocks:** Gameplay is continuous, and participants retain any tokens from incomplete suits across the entire game. However, the game is structured into blocks, and participants are informed that each block introduces a new probability distribution over the three suits. For example, in an 80-20 block, the probabilities might be 0.8 for *cat*, and 0.2 each for *hat* and *car*. The conditional probabilities of receiving a token given a card-action pairing in such a block are shown in Table 1 (see also Fig 1D). When a new block begins, the suit probabilities shift. For instance, in a 70-30 block, the dominant suit might now be hat, such that $\Pr(hat|\text{hat-related card}) = 0.7$, while $\Pr(cat|\text{cat-related card}) = 0.3$ and $\Pr(car|\text{car-related card}) = 0.3$ (see Fig 1E). Participants are explicitly informed about the block structure: each block lasts 30 rounds, different suits have different probabilities within a block, and these probabilities change when a new block begins.

## Experimental manipulations

**There is a dominant suit in each block.** We designed blocks in the game so that there is a dominant suit on each block (counterbalanced across all suit types), such that flipping its cards gives tokens with the highest probability (in Fig 1D, the *cat* suit is the dominant one with 0.8 probability). The other two suits have identically inferior token probabilities (both *hat* and *car* with 0.2 probability). We instructed the participants of this task manipulation, explaining that within each block there is a specific suit which has the highest chance of token collection. Pursuing the dominant suit type in each block gives participants the best chance at success in suit completion and we investigated how often participants chose the dominant option in a block in relation to their progress on various suits.

**Probability disparity between suits is modulated across blocks.** We also varied the probability disparity between the dominant and the inferior suits to ascertain how it modulates goal selection. We expected that selection of the dominant suit would be high in the high

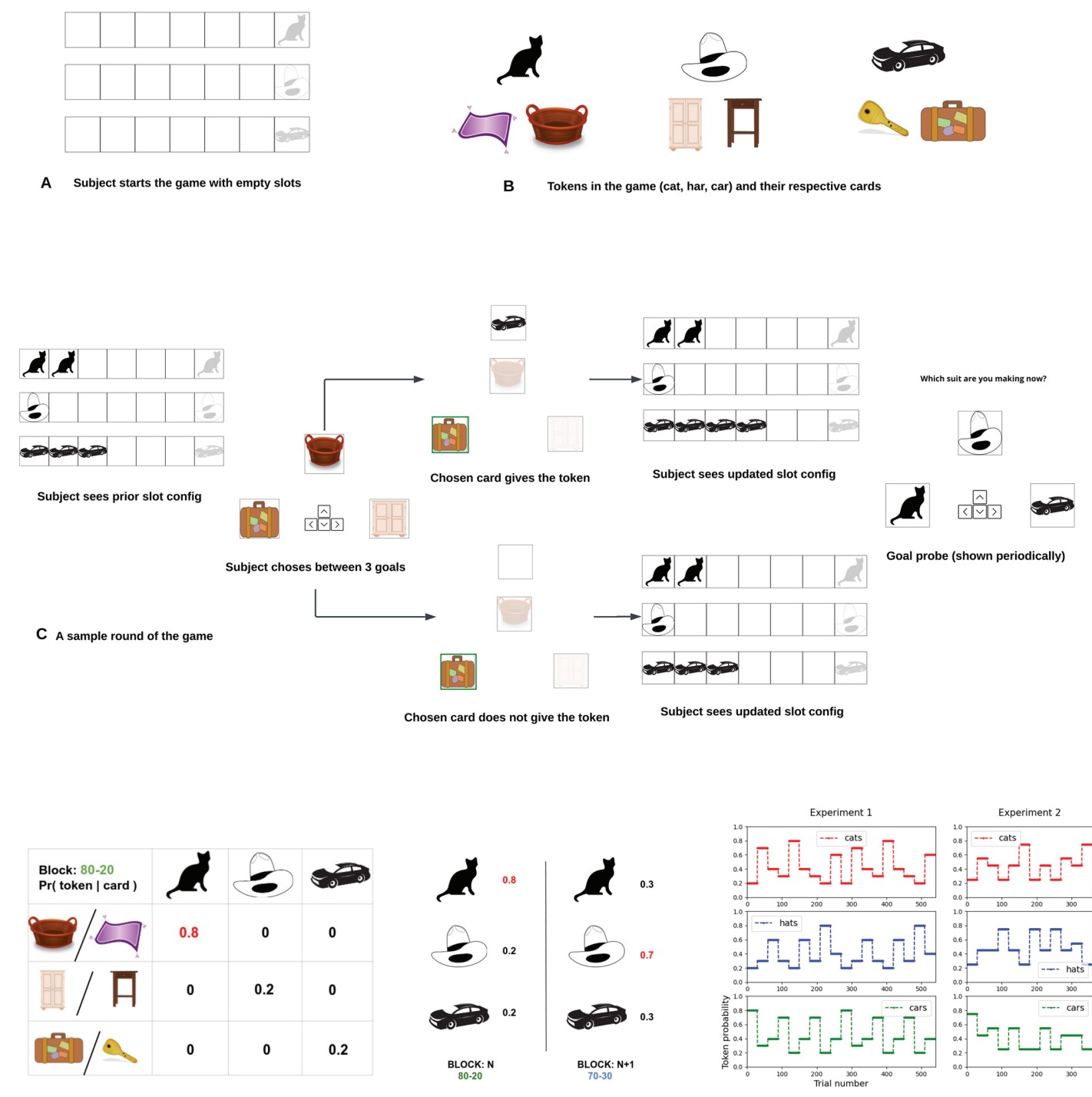

**Fig 1. Suits task. A.** Participants start with empty slots to collect tokens for each of the three available suits (*cat, hat, car*). **B.** Each suit type is mapped to two distinct cards which serve to stand in for the suits in the game. **C.** Panel shows a sample round of the task: 1. Participants see suit collection progress of all three goals. 2. Participants pick a card corresponding to one of the three goals. 3. Participants sees the outcome of their action 4. If a token is received, slots are updated. 5. Periodically, participants report the suit they are pursuing. **D. Conditional probabilities**. Pr(*token*|*card*) for different token-card combinations are shown for the 80–20 block. Selecting the *basket* or the *mat* card gives you the *cat* token with a high probability of 0.8, making *cat* the dominant suit in the block. **E. Block shifts** Each block has fixed probabilities with one dominant suit and two inferior suits of identical probabilities. The dominant suit is switched across adjacent blocks (*Cat* is most abundant in block N while *hat* is the most abundant in the next one). **F. Token probabilities.** Two experimental variants (experiment 1 and 2) with probability configurations specified for each suit. **Figure credits.** We used images from https://openclipart.org/ to generate the figures. See the Task design section in Methods for full image credits.

**Table 1. Pr(*token*|*card*) for the 80-20 block.** The dominant token is *cat* with Pr(*cat*|cat-related card) = 0.8

| Pr(*token*|*card*) | *cat* | *hat* | *car* |
|---|---|---|---|
| ***cat*-related card** | 0.8 | 0 | 0 |
| ***hat*-related card** | 0 | 0.2 | 0 |
| ***car*-related card** | 0 | 0 | 0.2 |

disparity blocks (where the probability difference between the dominant and the inferior suits is highest) and that this would be reflected in participants' goal selection patterns.

**The dominant suit is switched across adjacent blocks.** We implemented dominant token switches across blocks so as to enable us to investigate conflicts in goal selection between retrospective and prospective influences, as this is critical to assess how individuals change their choices of goal in the face of such a conflict. If a participant pursues a dominant suit in a block and is close to finishing it, and they encounter a block shift when the hitherto dominant suit is no longer dominant, then it is relatively more difficult to make further progress towards its target. A goal conflict arises in these situations, concerning whether to persist in the suit that is close to the target, or to switch over to the new dominant suit. Such scenarios occur frequently during the game, and the optimal strategy is to switch over to the dominant suit in the new block. We did not instruct the participants that the dominant token in one block becomes an inferior token in the adjacent block.

**Testing the effect of instructions.** We varied the instructions the participants received on the precise conditional probabilites across blocks. We investigated the extent to which knowledge of absolute probabilities of dominant and inferior tokens (without disclosing the identity of the dominant token) would influence goal selection in the game.

**Effect of non-uniform targets on goal selection.** We set non-uniform targets for the three suits to ascertain how that influences extended-valuation of the suits in relation to their progress and task-conditions.

We include four experiments to implement the above mentioned design manipulations. Experiments 1-3 have uniform targets for all suits, with each suit comprising of seven tokens of one kind. Experiment 1 has 18 blocks with 30 rounds each, with three different probability structures: 80-20, 70-30, 60-40. In the 80-20 block (say), the dominant suit is available with 80% chance and the two inferior tokens are available with 20% chance. Experiment 2 has 12 blocks (30 rounds each), with two different block types: 75-25 and 55-45. In both experiments 1 and 2, our instructions are limited in that, within each block, we explain that one specific suit has the highest probability, and that the probabilities change with each block switch. In experiment 3 (which is a repeat of experiment 1), we gave participants more information about their current block context. Before each block began, we specifically informed them that they are in an 80-20 context, without telling them which specific suit is available with 80% chance (more details on task design and instructions are elaborated in the Methods section). In experiment 4, we employed non-uniform targets for different suits and explored its impact on validating our computational hypothesis.

## Participants demonstrate a retrospective bias in temporally extended goal selection

We analyze goal selection patterns in human participants influenced by the current context (suit probabilities in a block) and slot configuration. In a given round, participants can choose to collect the dominant suit (with the highest probability of receiving tokens), or choose to

pursue the suit with the most progress (max progress suit) according to their slot configuration. In Fig 2A, we show the participants' probability of selecting cards related to the dominant suit, the max progress suit, and the third suit. Participants preferred to select either the dominant or the max progress suit across all conditions, picking the remaining suit well below the chance level. This indicates that the primary conflict in goal selection is between the accrued goal progress and the current rate of goal progress.

Moreover, the preference toward the dominant/max progress suit shows an interaction with the block condition. Participants show a higher preference for the dominant suit when the disparity between different suits is large (80-20/75-25 conditions) and a lower preference when the disparity is small (60–40/55–45 conditions), indicating sensitivity to the current rate of progress (repeated measures ANOVA; Experiment 1: $F(2, 86) = 20.23$, $p < 0.00001$; Experiment 2: $F(1, 49) = 17.32$, $p < 0.00001$). Likewise, preference toward the max progress suit increases with decreases in disparity between the suits (repeated measures ANOVA; Experiment 1: $F(2, 86) = 19.67$, $p < 0.00001$; Experiment 2: $F(1, 49) = 7.1$, $p < 0.01$). However, in all conditions, participants preferred to pick the maximum progress suit over the dominant suit, indicating an increased preference for the accrued progress over the current rate of progress.

Fig 2A provides a partial examination of the choice patterns, highlighting rounds where the dominant suit and the max progress suit are separate. However, in a fraction of rounds, the dominant suit is the same as the max-progress suit because maximum progress is made toward the dominant suit. In 62% of the rounds in Experiment 1 and 63% of the rounds in Experiment 2, the dominant suit is separate from the max progress suit, while in the rest of the rounds the two options converge. We noted that in the rounds where the dominant and the max progress suits are the same, both the accrued progress and the current rate of progress for the suit are high, mitigating conflict in goal selection. In those rounds, participants predominantly chose to pursue the dominant option (or the max progress option), picking the other two options occasionally and well below the chance level (see S1 Fig).

Limiting our analyses to the rounds where the dominant and max progress suits are separate (with a goal selection conflict), we further analyzed how progress differences between the two suits influence goal selection. Fig 2B shows the probability of choosing the max progress suit as a function of the difference in progress between the two suits. We show that participants' preference toward the max progress suit monotonically increases with the increasing progress difference between the suits. This preference is highest when the max progress suit is at 86% progress (6/7 tokens completed), and when the dominant suit has 0/7 tokens collected.

The pattern of increased preference for accrued progress in human participants is not necessarily suboptimal. If a suit is close to completion, it might be optimal to persist with it and get a reward sooner than to switch to the dominant suit that might have a high rate of progress but an equally high trials-to-completion estimate. In fact, humans are known to prefer sooner rewards over later ones, a tendency due to the incorporation of a discount factor in the estimation of the reward value incurred over a delay [27]. We investigated whether the goal selection patterns of participants are optimal under this assumption.

We constructed a prospective agent which prioritizes faster suit completion and reward accumulation. This agent, rather than optimizing for discounted returns accumulated at the end of the task, focuses on the immediate reward delivered at the end of suit completion. The agent thus places the highest value on goals that have the lowest estimates of

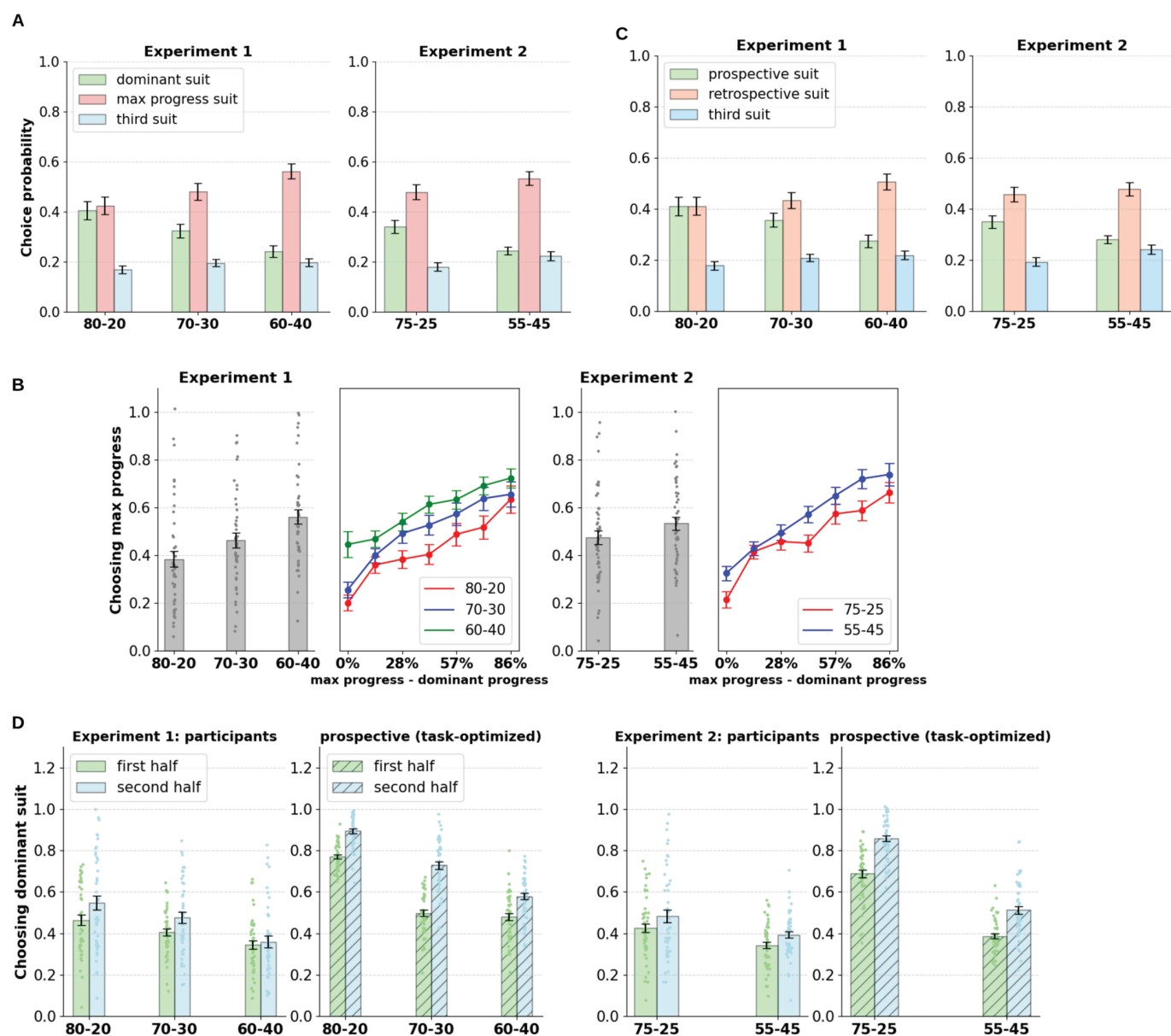

**Fig 2. A. Participants show a preference for the progress accrued over the current rate of progress.** Participants primarily chose between the dominant suit (one with the highest probability within the block) and the max progress suit (one which made greatest progress towards the target), while choosing the third option relatively infrequently. **B.** The proportion of times participants chose the max progress suit over the dominant suit (when the two diverge) is broken down by block type. The preference for the max progress option increases with the difference in progress between the dominant and max progress suits. **C.** Participants show a preference for the retrospective suit over the prospective suit (when the two choices diverge). Participants chose the retrospective suit (with the greatest progress) over the prospective suit (optimal choice prescribed the task-optimized prospective agent). **D. Learning effects within blocks.** Panel shows the dominant suit choice probability in the first and second halves of the block (15 rounds each). The proportion of dominant choices increases in the second half indicating learning effects within blocks. The choice patterns are contrasted with that of the task-optimized prospective agent, wherein, preference for the dominant suit is significantly lower in participants in both halves of the blocks, indicating suboptimal retrospective bias in goal selection.

trials-to-completion. To estimate values, the agent needs to maintain beliefs about the current probability estimates of various suits and roll out expectations into the future until the termination condition (collecting 7 tokens) is reached. We constructed the agent as a partially observable Markov decision process, with the current slot configuration as the observed state,

selection of cards as actions, and an estimate of the current suit probabilities as the belief state. The agent updates the belief about token-outcome probability of a suit using a delta learning rule:

$$M_g^{(t)} = M_g^{(t-1)} + \alpha(I_g^{(t)} - M_g^{(t-1)}) \tag{1}$$

where $M_g(t)$ is the belief at time $t$, $I_g(t)$ is the indicator function of the token outcome (1 if the token is received in a round), and $\alpha$ is the learning rate. Using the estimated probability of belief, the prospective agent rolls out expectations one step in the future for two scenarios: receiving a token with probability $M_g$ and moving one token closer to the target ($s_g^{(t)} + 1$); and not receiving the token with probability $1 - M_g$, and staying in the same slot state ($s_g^{(t)}$). The rolled-out future states are discounted by a factor of $\gamma$.

$$Q_g\big(s_g^{(t)}\big) = M_g\gamma Q_g\big(s_g^{(t)} + 1\big) + (1 - M_g)\gamma Q_g\big(s_g^{(t)}\big) \tag{2}$$

where $Q_g(t)$ is the value of the goal suit $g$, and $s_g(t)$ is the current slot state of suit $g$. Specifically, $s_g(t)$ is a counter for the progress of the goal, with the above recursive estimation terminating when the current state reaches the target (say, 7 tokens). Recursion is also terminated with a horizon of 20 steps.

The prospective agent, as defined above, can favor pursuing the maximum progress suit over the dominant suit. When the maximum progress suit is close to completion (say, 5 out of 7 tokens), the agent has to unroll fewer steps ahead to receive rewards compared to the dominant suit that is further away from the target (say, 2 out of 7 tokens). However, the prospective agent is sensitive to block conditions: When placed in the 80–20 block, the chance of collecting two additional tokens with a chance of 20% is lower than that of collecting five tokens with a chance of 80%, and the agent prefers to follow the dominant suit.

We optimized the parameters of the prospective agent (parameters and action selection process documented in the Computational Modeling section of Methods) for task performance (total number of suits completed). The optimal discount factor for this agent is $\gamma = 0.95$ for Experiment 1 and $\gamma = 0.93$ for Experiment 2. The prospective agent, thus optimized for the task, significantly outperforms the human participants (performance distinctions are elaborated in a later section).

For each slot configuration seen by human participants, we computed the response exhibited by the prospective agent and investigated the extent of the participants' deviation from this response. For generating the prospective responses, we assumed the true state of suit probabilities to be the agent's belief (not accounting for learning effects). We define the suit selected by the prospective agent as the prospective suit, and the suit with the maximum progress in a round as the retrospective suit. Fig 2C provides a revised analysis of Fig 2A, indicating the goal selection patterns to choose between the prospective suit, the retrospective suit, and the third suit (when the identities of the prospective and retrospective suit are different). The revised goal selection patterns show a trend similar to that shown in Fig 2A, with the preference for the retrospective suit being higher than that of the prospective suit in all conditions. Moreover, the prospective suit shows the same interaction profile with the block condition (repeated measures ANOVA: Experiment 1: $F(86) = 7.04$, $p < 0.001$; Experiment 2: $F(49) = 17.79$, $p < 0.0001$), and also with the retrospective suit (Experiment 1: $F(86) = 2.14$, $p = 0.12$; Experiment 2: $F(49) = 8.88$, $p < 0.005$). The proportion of rounds in which there is a conflict of goal selection in the current classification (prospective suit versus retrospective suit) is now reduced to 40% on average across all conditions (as opposed to 60% previously

in the max progress suit vs. dominant suit classification), indicating that a fraction of max progress choices are indeed prospective in nature after accounting for discounting effects. S1 Fig shows preferences when the prospective and the retrospective suit identities are the same (when the prospective agent prefers the maximum progress option), with participants predominantly choosing the prospective/retrospective option devoid of the goal selection conflict.

In summary, the patterns of goal selection in human participants reveal a retrospective bias for accrued progress in the task. Further quantifying the extent of this bias, Fig 2D shows the overall proportion of the choice of dominant suit in participants compared to that of the prospective agent optimized for the task. We also split the choices as occurring in the first and second halves of blocks (averaged over all blocks) to account for any learning effects. Both participants and the prospective agent show learning effects within blocks, with the proportion of dominant choice increasing in the latter half of the block. However, the dominant choice of the participants remains significantly lower than that of the prospective agent in both halves of the blocks, causing suboptimal human performance in the task (elaborated in a later section). Lower learning rates and increased discounting in participants could potentially explain suboptimality in task performance in humans, which we consider to be a potential explanation for the goal selection patterns seen in humans. This account is tested in later sections by fitting the prospective agent to the participants' choices. Furthermore, there are no significant effects of learning or performance changes between blocks. S3 Fig shows the proportion of dominant choice between blocks, where we find no significant changes in the selection effects as the task progresses.

## Switching patterns reveal greater persistence towards the retrospective suit

We further investigated temporal persistence patterns with regard to suit selection. Although overall goal selection is biased toward the retrospective suit, we further aimed to determine whether we could find evidence of a bias toward temporal persistence once a suit is selected. For this, we calculated the proportion of rounds in which participants switched away from the selected option in the subsequent round. We isolated these switches by suit type: separating rounds where they switched away from the prospective suit from those where they switched from the retrospective suit (similarly defined as in the previous section). Fig 3A shows the switch probabilities of the participants. We also separated the conditions where participants received a token in the previous round from those where they did not receive a token.

We found that participants made a significantly greater number of switches away from the prospective suit regardless of the token outcome in the previous round (paired-sample t-test; Experiment 1: $t(86) = -2.22$, $p < 0.01$; Experiment 2: $t(96) = -2.79$, $p < 0.001$). We found that the proportion of prospective switches interacts with the block types (repeated measures ANOVA; Experiment 1: $F(2, 144) = 12.56$, $p < 0.0001$; Experiment 2: $F(1, 49) = 19.55$, $p < 0.0001$). Participants are most likely to persist with the prospective suit in the high-disparity blocks (80–20/75–25), where the adaptive value of persisting with the prospective token is maximal, compared to the low-disparity blocks (60–40/55–25). However, overall levels of persistence with the prospective suit are lower than those of the retrospective suit.

Retrospective switch propensies are also sensitive to the relative progress differences between the two suits and are lower when the difference is high. Fig 3B shows switching proportions away from prospective and retrospective suits split by high ($> 50\%$) and low ($\leq 50\%$) progress difference between the two suits. Participants demonstrate increasing

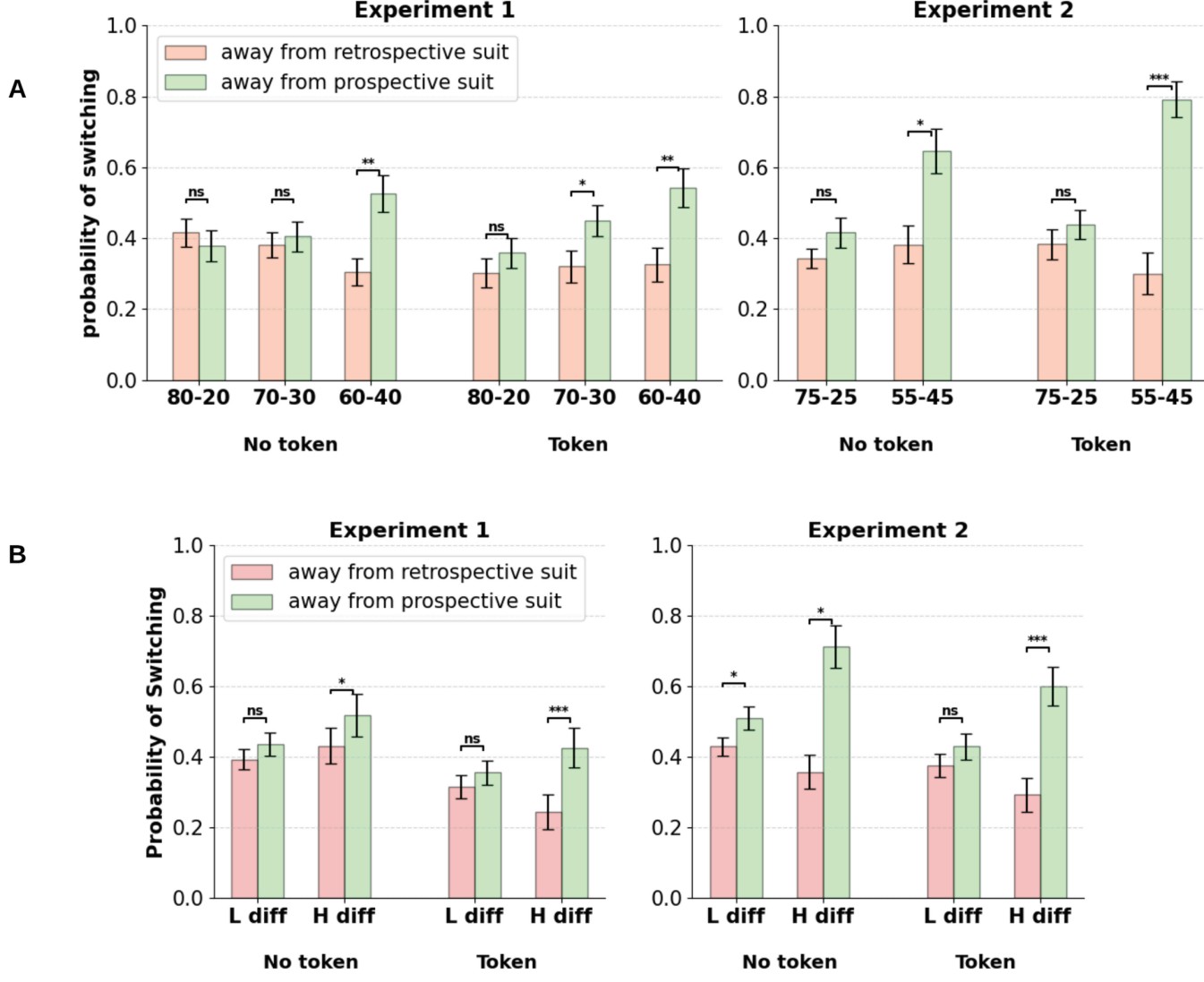

**Fig 3. A. Participants show greater persistence towards the retrospective suit.** Participants make more switches away from the prospective option than the retrospective option. Switches are split by suit type (prospective/retrospective), block condition, and whether the round elicits a token or not. Significance markers indicate paired-sample t-tests ($*: p < 0.01$, $**: p < 0.001$, $***: p < 0.0001$). **B. Switches split by progress differences between the suits.** H diff, L diff correspond to higher and lower progress difference between the suits. Increased persistence is observed with the retrospective suit when progress difference between the suits is high.

levels of commitment toward the retrospective suit with its increasing distance (progress) from the prospective suit. This is reflected in lowered switching from the retrospective suit and increased switching from the prospective suit. Increased commitment toward the retrospective choice is thus evidenced not only by resistance to switching from the choice but also by increased rates of revisiting it after having switched away from it.

This points out an interesting facet of goal persistence which can be dissociated from mere stickiness in choice driven by task-switching costs. The persistence in goal selection that we describe here tends to involve a tendency to revisit the retrospective option even after it has been previously switched away from, whereas choice stickiness typically refers to a tendency to refrain from switching away from a particular action in the first place. Thus, accrued

progress in a particular goal seems to exert a retrospective pull on subsequent decisions, clearly highlighting persistent retrospective influences on goal selection.

## Human performance falls short of the prospective agent optimizing for discounted future rewards

We showed evidence that human participants exhibit a bias towards selecting and persisting with the retrospective suit. Here, we analyze how bias translates into suboptimal performance in the task.

We compare the performance of human participants with that of the task-optimized prospective agent. For further contrast, we also constructed a retrospective agent that always prefers the suit that made the most progress toward the target through the following value estimation process.

$$Q_g\left(s_g^{(t)}\right) = \gamma Q_g\left(s_g^{(t)} + 1\right) \tag{3}$$

The retrospective agent also applies a recursive estimate similar to that of the prospective agent to discount future rewards (with a factor $\gamma$), but without incorporating the belief state about the current suit probabilities (Equation 3). In effect, the agent always values the suit that is closest to the target.

We incorporated the same goal selection algorithm as that of the prospective agent (see Computational Modeling section in Methods), and likewise, the parameters of the retrospective agent are optimized for task performance.

In general, human performance does not match the prospective agent in both experiments (Experiment 1: $t(86) = -9.98$, $p < 10^{-15}$; Experiment 2: $t(98) = -9.60$, $p < 10^{-15}$), as shown in Fig 4A. The contrast is particularly striking on conditions where there is a high disparity in the probabilities of token outcome (80-20 and 75-25 blocks). However, the performance contrast is not significant in the blocks where the benefit of prospecting is minimal (60-40 and 55-45 blocks). The prospective agent shows a significant interaction of performance with the block type (Experiment 1: $F(2, 86) = 49.85$, $p < 10^{-14}$; Experiment 2: $F(1, 49) = 147.8$, $p < 10^{-15}$), whereas human agents show a diminished yet significant interaction with block type in Experiment 1 ($F(2, 86) = 3.81$, $p < 0.05$) and no significant interaction in Experiment 2. Human performance beats the retrospective agent marginally in Experiment 1 ($t(86) = 3.28$, $p < 0.01$), while the difference in performance distributions is not significant in Experiment 2. Task performance mirrors the heterogeneity evident in participants' retrospective choices, with only a few participants attaining performance close to that of the optimal prospective agent.

Fig 4B compares the proportion of choice of the max progress option of the two agents with that of the human participants. The retrospective agent, in the spirit of its formulation, demonstrates the highest preference for accrued progress, whereas the prospective agent falls at the other extreme; human participants fall part-way between the two agents, indicating an inflation of retrospective choice beyond what is optimal for the task.

## Explicit instruction about contingencies within a block alleviates but does not eliminate the inflation of retrospective choice

Although we explicitly designed the block conditions so that there is a dominant suit, it is likely that participants did not anticipate how effective the pursuit of the prospective suit over the retrospective choice would be, thus reducing the exploration of alternative suits. If

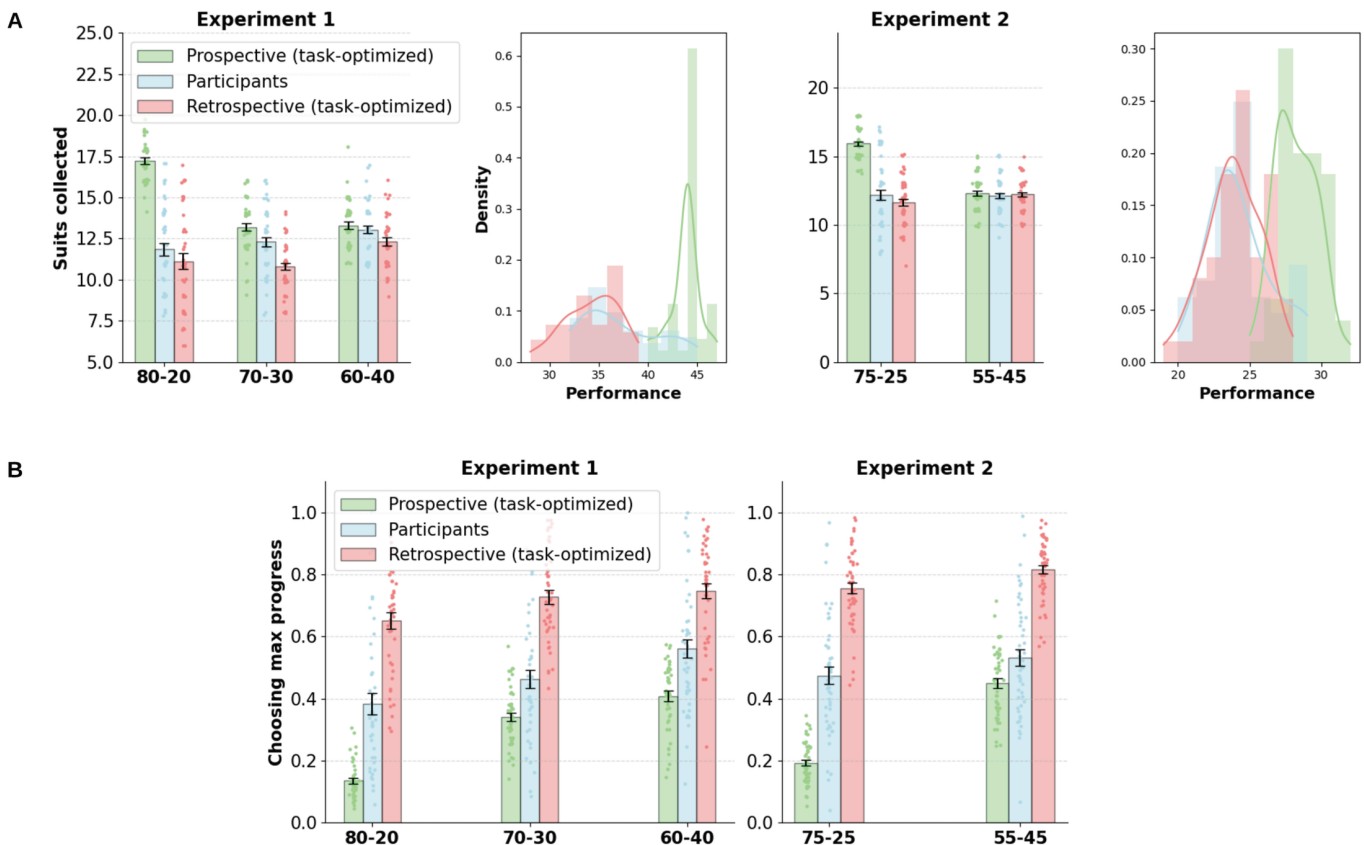

**Fig 4. A, B. Human performance contrasted against task-optimized prospective and retrospective agents.** Human behavior significantly falls short of optimal prospective agent and marginally beats the retrospective agent in task performance (total number of suits collected). Agents' parameters are optimized for performance in the task. Performance is broken down by task condition and the distribution of performance is shown. **B. Preference towards accrued progress shown by the two agents contrasted with human participants.** The retrospective agent has the highest proportion of retrospective choice and the prospective agent the lowest, while human retrospective preferences lies between these extremes.

information regarding the abundance of the prospective suit is made explicit, will it lower retrospective influences on the task?

In Experiments 1 and 2, we simply alerted the participant when a block shift occurred. We now consider a variant of the first experiment, which we call Experiment 3, where we gave explicit instructions to participants about the contingencies within each block. Before the start of the game, we instructed the participants on the types of blocks present in the task and also alerted the participants regarding the available contingencies at the beginning of each block. For example, we told them that they are in an 80-20 block condition. We nevertheless refrained from telling the participants which specific suit is the dominant one in a block to avoid explicitly priming a goal switch.

Fig 5A plots the performance of the task with explicit contingency information alongside the variant without contingency information, compared to the prospective agent. Task performance improved with contingency (independent samples t-test: $t(106) = -2.00$, $p < 0.05$) - evident in the 80-20 block condition - yet it is still significantly short of the prospective agent.

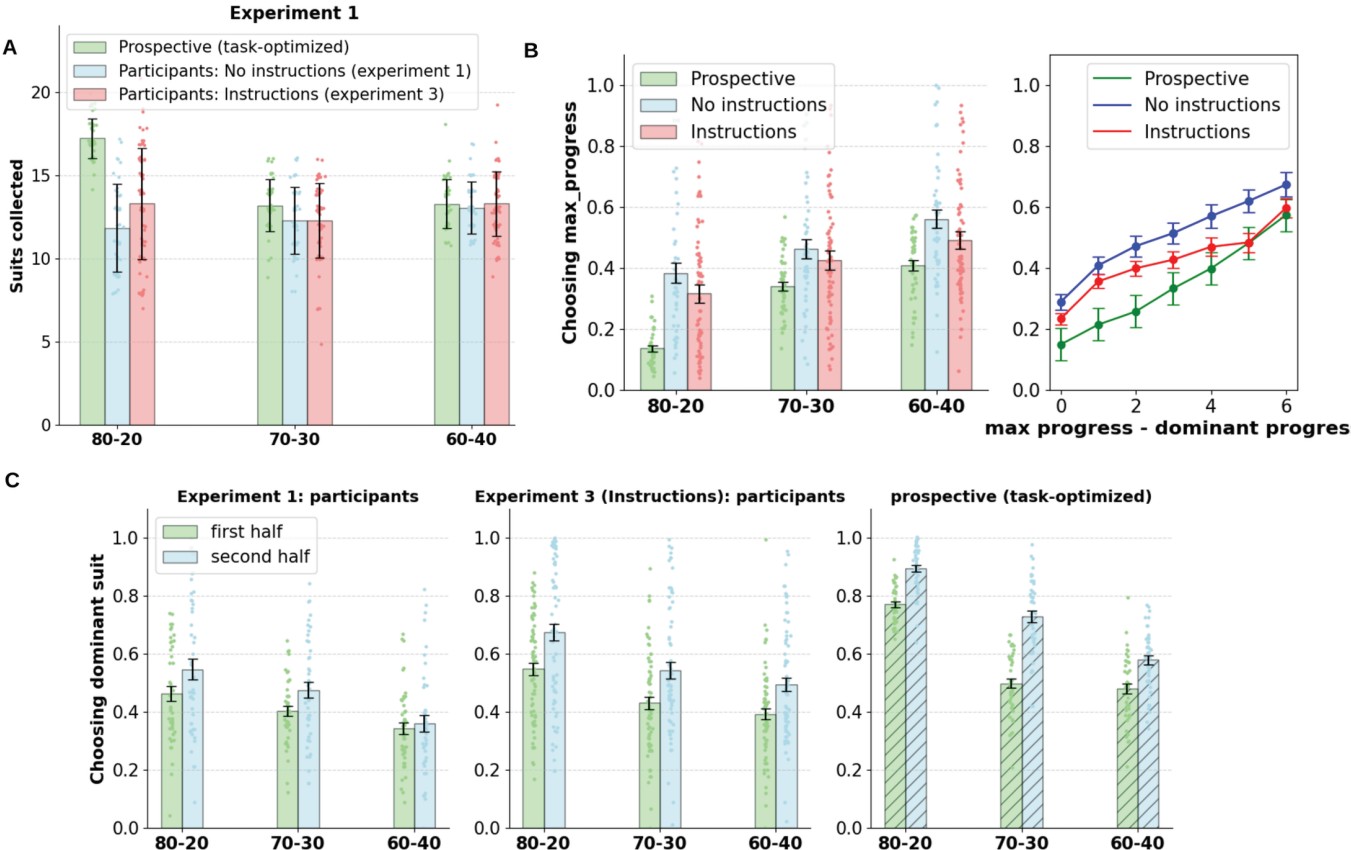

**Fig 5. A, B. Effects on explicit instruction about prospects of alternative goals on behavior. A.** Task performance improves with instructions yet significantly falls short of the optimal prospective agent. **B.** Retrospective choice proportion decreases with instructions yet is significantly inflated when contrasted with the prospective agent. **C. Learning effects within blocks.** Panel shows the dominant suit choice probability in the first and second halves of the block (15 rounds each) in the presence and absence of explicit instructions. Preference towards the dominant choice increases in both halves of the blocks indicating a shift towards optimal goal selection in the presence of instructions.

Fig 5B shows the contrast in the choice proportion of the max progress suit found in versions with and without instructions. Explicit instructions reduced the retrospective influence on goal selection, yet did not eliminate its effect relative to an optimal prospective agent.

Fig 5C contrasts the proportion of choice of dominant suit in the first and second halves of the blocks in the version with instructions (Experiment 3), without instructions (Experiment 1), and the prospective agent optimized for the task. Although the proportion of dominant choices increased in the version with instructions, it remained significantly lower than the task-optimized prospective agent, indicating suboptimal goal selection patterns in human participants even in the presence of explicit instructions on contextual goal prospects.

## Momentum as a key computational variable driving the persistence of temporally extended goals

We previously described an optimal prospective agent as a benchmark for human performance. A potential account for suboptimal human performance could be that humans incorporate token-outcome contingencies of the suits inefficiently, or alternatively, humans could be using an entirely different mechanism to estimate the likelihood of achieving future goals.

As an alternative account, we formalized momentum as a measure of the goal value that drives persistence and is defined as the product of progress made toward the goal and its speed toward the goal.

$$\text{goal momentum} = \text{progress accrued} \times \text{current rate of reinforcement} \tag{4}$$

In the case of multiple goals with different rewards, the above value can be scaled appropriately with the associated rewards.

This equation is similar in essence to the construct of 'momentum', defined as the product of mass and velocity in Newtonian physics, which explains the resistance of an object to change. The larger the momentum, the greater its resistance to change in value. The analogy of goal momentum with the physical interpretation of momentum goes only as far as it comprises two factors: accrued progress (analogous to behavioral mass) and the current rate of progress (analogous to velocity), which together create resistance to change in valuation. A longer history of reinforcement with an option (accumulated behavioral mass) and a higher current rate of reinforcement both increase momentum and facilitate persistence with an option. This analogy was previously used in the definition of behavioral momentum [9], where the rate at which the free operant responds in the presence of a stimulus is likened to the velocity of a moving body, and the resistance to change of the response is likened to its inertial mass. We further extend this metaphor to include extended goals, where the momentum of the goal in reaching its desired setpoint determines its value.

Momentum, in the purview of goal pursuit, is an estimate of the drive toward a specific goal that is dependent on both the history of reinforcement (previous progress) and the current rate of reinforcement. We define the progress accrued in a fractional format (say 40%) that facilitates a relative comparison between goals with distinct objectives. Formulated in this way, the progress accrued is also the inverse of proximity to the target. From this perspective, goals can be viewed as setpoints in the distant future, and the value of the goal is its momentum in reaching the setpoint.

With the above definition, our aim was to capture the two effects that we found in behavior. As progress accrued in a suit increases, the resistance to switch away from the suit is higher, and as the speed of progress reduces (as a result of shifting token contingencies), the resistance to switching is lowered. Finding parallels to the definition of physics, we equate the progress accrued as a measure of mass causing inertial effects, in line with the endowment effects seen with physical objects in behavioral economics [28]. While progress accrued increases inertia in switching, a sufficiently lowered speed of progress can lower momentum, and thus facilitate switching away from the option.

**TD-momentum** We use the aforementioned conceptualization of momentum, a product of progress accrued and the speed of progress, as a measure of the goal value.

$$Q_g\left(s_g^{(t)} = m^{(t)}\right) = v_g^{(t)} m^{(t)} + b_g^{(t)} \tag{5}$$

where $Q_g$ is the estimate of the goal value of a suit $g$ in a specific slot configuration state $s_g^{(t)}$, $m$ is the fractional progress made toward the goal in a given instant $t$, $v_g$ is the parameter for the current rate of progress and $b_g$ is a goal-specific bias for $g$.

We formulate an algorithm, TD-momentum, to facilitate updating of momentum estimates for different goals. Here, the product of the progress accrued and the rate of progress serve as a function approximator of the goal value, with speed as a free parameter and progress as an indicator of the state of the goal [29]. The estimation of speed is done through the temporal difference learning rule, with its value updated through gradient descent on the prediction error of the expectation of receiving a token in the next round (and progressing one step ahead).

The temporal difference error ($\delta$) is estimated as follows:

$$\delta = \gamma Q_g\left(s_g^{(t+1)}\right) - Q_g\left(s_g^{(t)}\right) \tag{6}$$

The TD momentum parameters $v_g$ and $b_g$ are updated with a learning rate $\alpha$ through a gradient descent in the value function $Q_g$.

$$v_g^{(t+1)} = v_g^{(t)} + \alpha\delta\frac{dQ_g^{(t)}}{dv_g^{(t)}} \tag{7}$$

$$= v_g^{(t)} + \alpha\delta m^{(t)} \tag{8}$$

$$b_g^{(t+1)} = b_g^{(t)} + \alpha\delta \tag{9}$$

The goal-specific bias term ($b_g$) is included to ensure that the goal value is updated even when progress is at $m = 0$. When one chooses the goal when its progress is zero and repeatedly fails to achieve the outcome, its value is depreciated through the bias term. The bias term also ensures that the goal value is retained when the goal target is accomplished (upon which the progress $m$ resets to zero), and can increase the chance of choosing the same goal again even though the value component of the term $v_g m$ is zero.

Here are the key features of momentum computations for goal valuation.

1. **The relation between goal momentum and fractional progress is modulated by the discount factor, the step size of progress, and the current rate of progress**

   If no unit progress is made in a round, $s_g^{(t+1)} = s_g^{(t)} = m$, $\delta$ is always negative for $\gamma < 1$.

   If unit progress ($\eta$) is made in the next round, $s_g^{(t+1)} = m + \eta$. Assuming no goal-specific bias,

$$\delta = \gamma Q_g\left(s_g^{(t+1)}\right) - Q_g\left(s_g^{(t)}\right) \tag{10}$$

$$= \gamma(m+\eta)v_g - mv_g \tag{11}$$

$$> 0 \iff \gamma > \frac{m}{m+\eta} \tag{12}$$

   With unit progress ($\eta$), prediction errors ($\delta$) are positive if and only if the condition $\gamma > \frac{m}{m+\eta}$ is satisfied (Equation 12).

   For example, given $\gamma = 0.8$, the unit progress from $m^{(t)} = 0.2$ to $m^{(t+1)} = 0.25$ will yield a positive prediction error, while the unit progress from $m^{(t)} = 0.9$ to $m^{(t+1)} = 0.95$ produces a negative prediction error. Therefore, for a fixed $\gamma$, a small enough unit of progress can result in negative speed updates when the fractional progress ($m$) is close

to 1. This accounts for the phenomenon of diminishing marginal utility [30], whose main feature is that the marginal utility of a unit increase in a good decreases as its quantity increases.

When no progress is made in a round ($s_g^{(t+1)} = s_g^{(t)}$), $\delta = \gamma Q_g\left(s_g^{(t+1)}\right) - Q_g\left(s_g^{(t)}\right) < 0$, and updates for the speed of progress are always negative. On the other hand, if unit progress ($m \rightarrow m + \eta$) made in a round, the changes to the speed term are positive, but only if $\gamma > \frac{m}{m+\eta}$. Thus, TD-momentum value estimates do not need to increase monotonically with progress; their relation to progress is determined by the discount factor $\gamma$, the step size of unit progress $\eta$, and the current estimate of the rate of progress ($v_g$).

Fig 6A demonstrates this feature of momentum computations. We simulated TD-momentum value estimates for several discount factors, with a step size of unit progress of 0.14, and probability of making unit progress as 0.6 (binary 0s and 1s drawn with probabilities 0.4 and 0.6, respectively). We note that for $\gamma < 0.9$, TD-momentum exhibits diminished value with increasing progress when the fractional progress is close to one.

However, for most participants in our task, the TD-momentum $\gamma$ estimates through behavioral fits indicate a discount factor $> 0.9$, so nonmonotonic behavior with increasing progress is not observed in the task. However, we expect that with sufficiently lowered step sizes of progress, the TD-momentum can predict diminished utility closer to goal completion.

2. **Goal momentum provides stable value estimates against fluctuations in reinforcements** We contrast the TD-momentum value estimates with that of the prospective agent in Fig 6B. In the simulation, we introduced a change in the probability of unit progress from 0.6 to 0.2 after 15 rounds of the task. We note that the value estimates from the prospective agent exhibit fluctuations corresponding to individual reinforcements of unit progress, whereas momentum builds up value steadily and gradually decays when encountered with a shift in probability. We believe that this retention of stable value estimates enables persistence beyond a context shift, which could facilitate sustained commitment toward long-term goals.

**Alternative models.**   We constructed several alternative models to enable us to test several alternative hypotheses to the momentum explanation in accounting for human behavior on the task.

**Perturbed prospection.** Firstly, we devised parameterized variants of the prospective agent that learns the token outcome probabilities inefficiently, employing both symmetric and asymmetric learning rates [31] for positive and negative prediction errors. We expected that a prospective agent acting on noisy/inertial/biased probabilities could account for suboptimality in choices. We experimented with several variants of the prospective model and observed that the simple variant with symmetric learning rates captures the data better than the others (see S6 Fig and S1 Appendix).

**Hybrid.** Secondly, we conceived a hybrid model that incorporates weighted contributions from prospective and retrospective agents, which could potentially account for retrospective bias in goal switching.

**TD-persistence.** Lastly, we constructed an agent – TD persistence – that learns the token outcome expectations of each goal using a temporal difference learning rule and acts guided by those expectations, modulated by persistence parameters in action policy. As we instructed the participants that every block has different probabilities, a dominant strategy would be to estimate which goal has the highest token-outcome expectation and persevere with it until the end of the block. We expected the influence of this strategy to be greater in the experimental

variant (Experiment 3), where we provided explicit instructions about the type of their current block. TD persistence, having an optimal performance similar to that of the prospective agent (see S7 Fig), is one of the better strategies in the task. However, by its very nature, it cannot reproduce either the temporal patterns of retrospective bias or the switching trends seen in behavior.

We incorporated a goal perseveration parameter in the common action selection policy (see Methods)—a choice kernel that increments each time a goal is selected and decays each time it is not—so model comparisons elicit differences beyond any choice persistence or putative "sunk cost" mechanisms.

We ensured that the competing hypotheses elucidated above are individually recoverable (see Methods on parameter and model recovery, also S4 Fig and S5 Fig).

**Comparison to simpler reinforcement learning models.** We also considered the possibility that participants did not play the game within the framework of extended goals, but rather selected goals by treating them as individual unconnected events in which one simply receives or does not receive a token reward. In such a scenario, behavior could be captured by a simple reinforcement learning (RL) model utilizing a standard delta learning rule such as in the trial-based Rescorla-Wagner model [32]. This simplified RL model could potentially predict an increase in retrospective choice due to prior reinforcements increasing expected values of tokens. However, participants were alerted during block shifts to ensure minimal carryover of such expectations of token probabilities. Nevertheless, to rule out this possibility, we fit participants' choices to this simplified RL model [33] (while also including a choice kernel), and we generated simulated data from the fitted parameters (see Methods). The model-generated data capture neither the preference for accrued progress nor the sensitivity to relative progress differences between the suits (see S13 Fig). That is to be expected, because this simplified RL model has no notion of commitment toward extended goals, supporting our presumption that

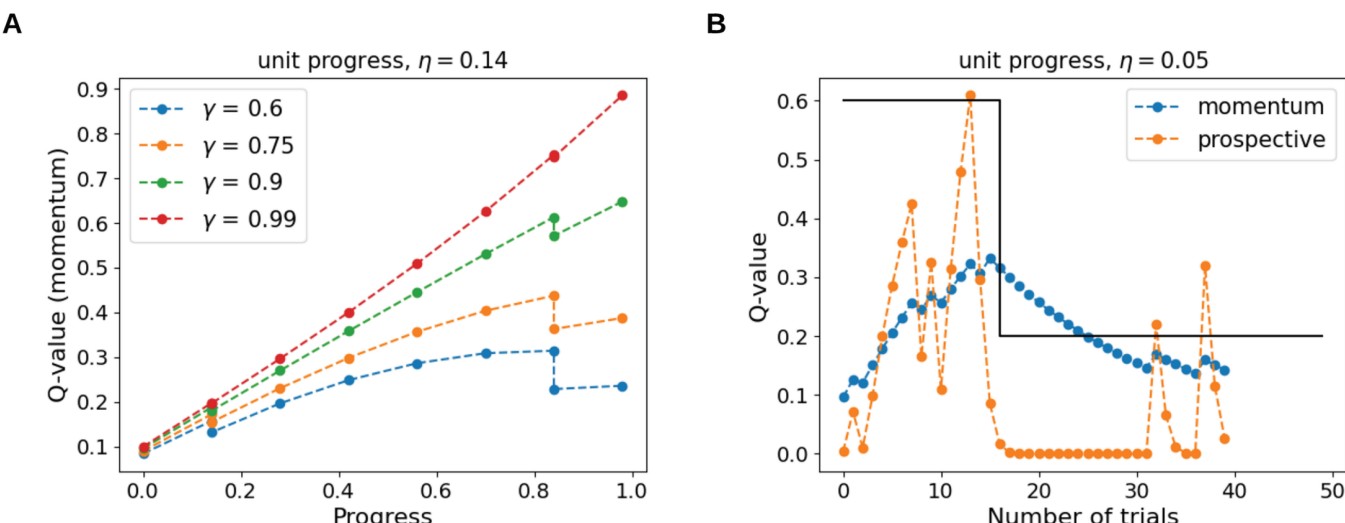

**Fig 6. A, B. Features of the TD-momentum algorithm. A.** The relation of TD-momentum value estimates with progress is determined by the discount factor, step size of progress, and the probability of making unit progress. In the simulations, the probability of unit progress is 0.6, and the step size of unit progress is 0.14. **B.** Comparison between the value computations from prospective and TD-momentum algorithms. The black line indicates the shift in probability of unit progress from 0.6 to 0.2 after 15 trials. TD-momentum provides stable value estimates, as opposed to the prospective model which exhibits fluctuations in response to individual reinforcements. Step size of unit progress is 0.05, learning rate is 0.4, and $\gamma$ is 0.9 for both algorithms.

the reinforcement schedule imposed by extended goals inflates retrospective choice over and beyond the predictions of such a simplified RL model.

**Time-inconsistent discounting.** We employed exponential discounting to design the task-optimized prospective agent. However, given that human behavior is better accounted for by hyperbolic discounting [27], we also implemented a prospective agent with hyperbolic discounting. We estimated trials to completion (delay to reward $D$) for each goal by rolling out the results into the future and used the hyperbolic discounting factor ($\gamma = \frac{1}{1+kD}$, $k \in [0,1]$) to discount the rewards. However, we found no significant difference in behavioral fits between the variants of the prospective agent with exponential and hyperbolic discounting (see S6 Fig).

**The TD-momentum model offers a better fit to human choice data.** We fit each of the above models to the participants' choices of stay/switch choices from actions and goals reported from the previous round and compared the models using the Akaike Information Criterion (AIC) and the Bayesian Information Criterion (BIC) metrics.

$$AIC = 2k - 2\ln(\hat{L})$$

$$BIC = k\ln(n) - 2\ln(\hat{L})$$

Here, $n$ is the number of data points, $k$ is the number of parameters in the model, and $\hat{L}$ is the maximum value of the likelihood function. The lower the AIC/BIC metrics, the better the fit of the model. Moreover, we computed a 3-fold cross-validation (CV) metric for each participant by fitting the model with two-thirds of the task blocks and computing the negative log-likelihood on the held out one-third data (mean metric computed over 30 random splits of the data). Fig 7A and 7B show that the TD-momentum offers a better account of the data according to the AIC, BIC, and cross-validation metrics. Beyond just aggregate likelihood comparisons between models, we examined whether TD-momentum provided superior fits consistently across individual subjects [31]. We performed paired sample t-tests between AIC, BIC, and CV metrics, and TD-momentum consistently offers better fits between participants in experiments 1 and 2 compared to other models.

Fig 7C shows a comparison between the behavioral predictions of the different models in the presence of explicit instructions on the types of contingency (experiment 3), contrasting with that of the variant without instructions (experiment 1). In both versions, TD-momentum offers a better fit to the data. However, we observed an increased contribution of TD-persistence at the level of individual participants in the version with explicit instructions. Fig 7D shows the difference in BIC between the two algorithms in the individual participants who contrast the two variants. In experiment 1, TD-momentum offers a better fit for 72% of the participant pool (loglikelihood ratio test with $p = 0.05$), whereas the number drops to 58% in experiment 3. We predicted that 42% of participants who were better accounted for by the TD-persistence model in the version with instructions would perform better on the task than those of participants accounted for by the TD-momentum model. In fact, there was a significant difference in performance between the two groups (independent samples t-test: t(63)=$-3.69$, $p < 10^{-3}$). Retrospective bias patterns in the two participant groups (TD-momentum and TD-persistence) follow the same patterns as those obtained from simulations of respective algorithms (see S8 Fig).

**The TD-momentum model reproduces the behavioral effects of the aggregated participant pool.** Using the parameters derived from individual participant fits of TD-momentum, we generated synthetic data imitating the participant pool to validate whether the model can replicate key behavioral effects that capture retrospective influences in experiments 1 and

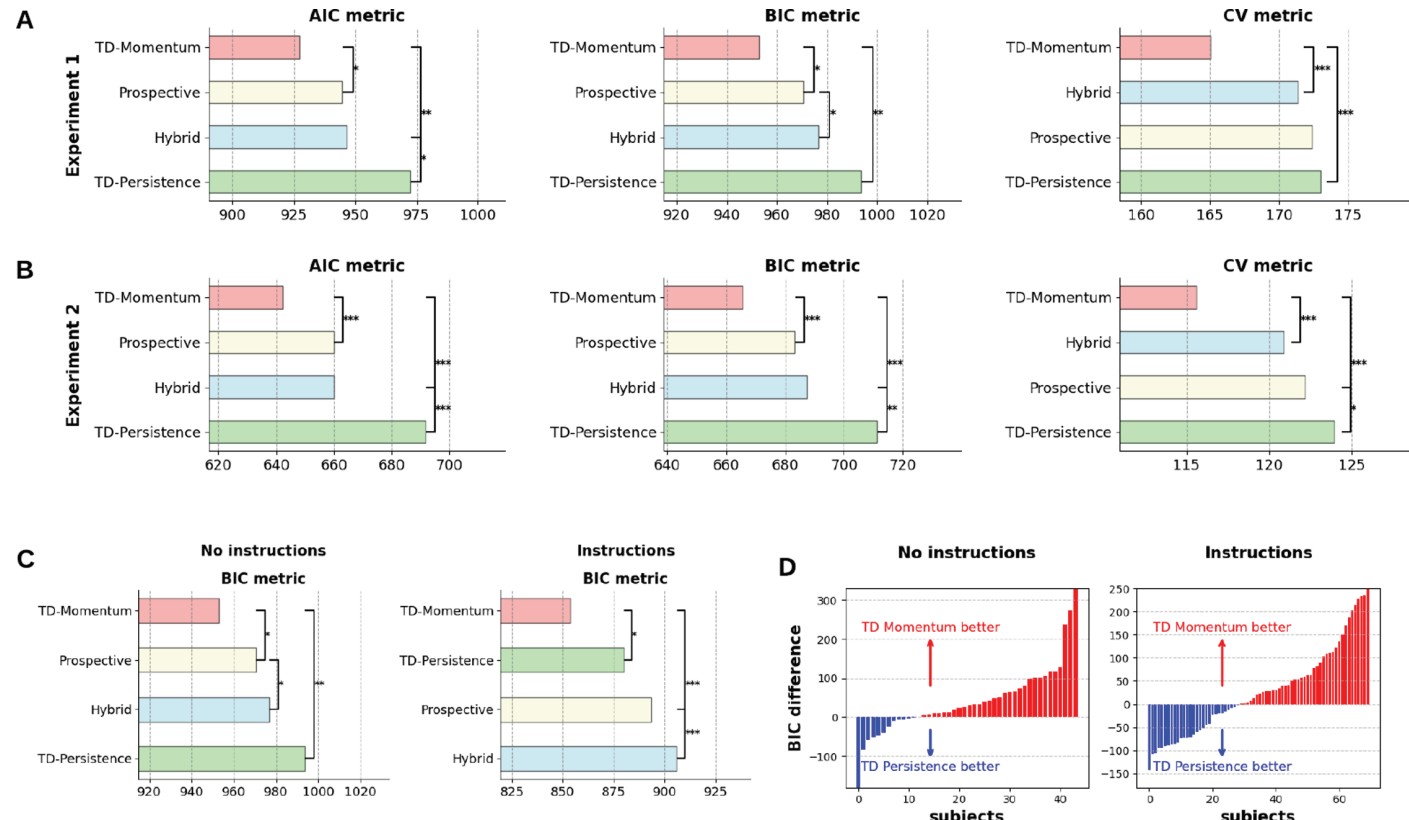

**Fig 7. A, B. Model comparisons.** TD-momentum accounts for the switching choices in the task better than the other prospective, hybrid, and TD-persistence models as per AIC, BIC, and 3-fold cross validation metrics in experiments 1 and 2. Mean and standard error bars of the metrics shown in the figure. Significance markers correspond to two-tailed paired sample $t$-tests ($*: p < 0.01$). **C.** TD-momentum also offers the best account of behavior in the variant with explicit instructions. **D.** Incidence of the competing hypothesis of TD-persistence increases in the version with instructions.

2. One sample of synthetic data per real participant is generated using the fitted parameters, and the aggregate choice patterns of the population of model-simulated participants are contrasted with those of the real participants. Simulated participant data reproduces interaction effects of preference for accrued progress with the block type and also captures the pattern of increase in preference for accrued progress with increasing progress difference between the maximum progress suit and the dominant suit (see Fig 8A and 8B). The simulated data also captures the trend of greater persistence in favor of the retrospective suit over the prospective one in switching patterns (see Fig 8C and 8D). The prospective and hybrid agent also captures the behavioral trends reasonably well (see S10 Fig and S11 Fig), while the TD-persistence agent fails to capture them (see S12 Fig).

We also plotted how individually fitted model parameters correlate with participants' task measures. The momentum learning rate correlates with the performance in the task (Experiment 1: R=0.78, $p < 10^{-9}$; Experiment 2: R=0.45, $p < 0.0008$; Experiment 3: R = 0.68, $p < 10^{-10}$), that is, participants who exhibited greater learning of the speed of progress of different goals collected more suits in the task. The cost of switching is correlated with the number of action switches made by the participant (Experiment 1: R=-0.78, $p < 10^{-9}$; Experiment 2: R=-0.81, $p < 10^{-12}$; Experiment 3: R = −0.62, $p < 0.001$), that is, participants with

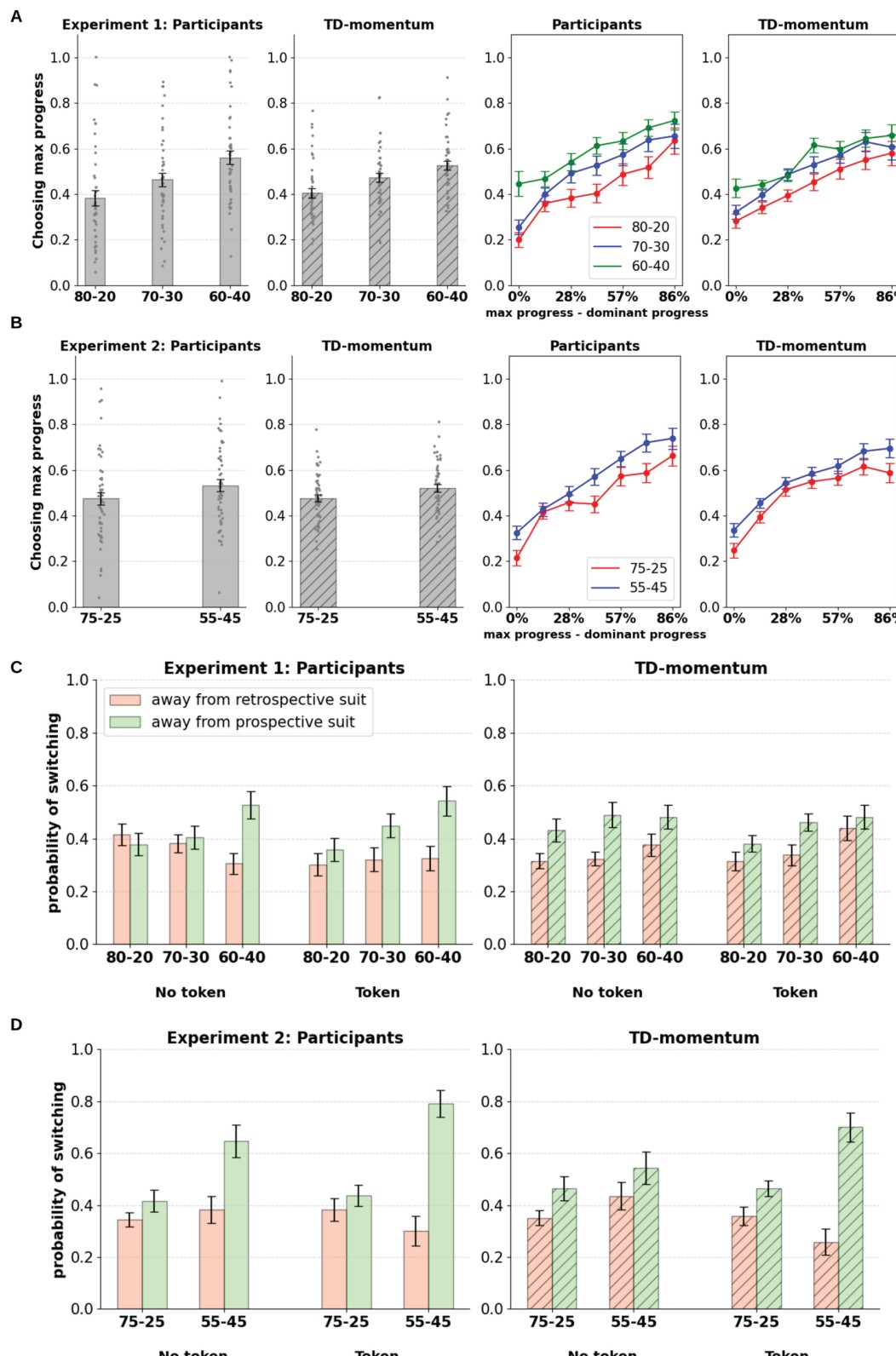

**Fig 8. TD-momentum: model predictions** TD-momentum agent captures general trends in preferences towards maximum progress suit and switching patterns in the task.

lower switching costs made more switches under all conditions. Lastly, the softmax temperature parameter correlates with the preference for accrued progress (Experiment 1: R=0.73, $p < 10^{-7}$; Experiment 2: R=0.75, $p < 10^{-10}$; Experiment 3: R = 0.60, $p < 10^{-7}$), that is, the more the participant relies on the value disparity between the goals generated by the momentum model, the greater is their preference for accrued progress. TD-momentum also reproduced participants' learning effects within a block and captured the increase in dominant suit choice proportions as the block progresses (see S9 Fig).

**Non uniform targets for goals delineate distinctions between TD-momentum and prospective agents.** Although the TD-momentum model outperforms all other models in explaining the likelihood of participants' choices in the task, three of the agents, TD-momentum, prospective, and hybrid, qualitatively reproduce key behavioral signatures in the task. In order to more definitively differentiate between which of these models best explains human behavior, we turned to an additional experiment.

A key distinction between the TD-momentum and the prospective agent is that the former relies on a measure of fractional progress, and the latter relies on the absolute distance to the target. In experiments 1 and 2, both measures converge, as targets for all suits are the same with no distinctions between fractional progress and absolute proximity to the target.

In experiment 4, we introduce variable target token lengths for the suits. The suit *cat* can be completed by collecting 4 cat tokens while the suits *hat* and *car* can be completed by collecting 6 and 8 tokens, respectively. Thus, if the *cat* suit has 2/4 tokens completed and the car suit has 6/8 tokens completed, the prospective agent has to unroll two steps ahead to predict rewards for both suits, however, TD-momentum considers that the *cat* suit is 50% done and that the *car* suit is 75% done. Thus, this design emphasizes the distinctions between the two algorithms.

Fig 9A shows the slot configuration for experiment 4. We adjusted the token probabilities so that there is one dominant prospective suit in each block and so that the dominant prospective suit is counterbalanced across suits. We introduced high- and low- disparity block conditions (*H disp* and *L disp*). In both conditions, the dominant suit takes 8 rounds (on average) to complete, while in the *H disp* and *L disp* conditions, the two inferior tokens take 15 and 12 rounds (on average) to complete. Fig 9B shows the participants' goal selection probabilities divided by the dominant suit condition. The participants selected the *cat* suit with the highest chance in the *cat* dominant blocks, etc. for the other suits, indicating sensitivity to token contingencies. Participants also showed sensitivity to relative distinctions between suits with a lower proportion of dominant suit choice in the low-disparity blocks compared to the high-disparity blocks (Fig 9C). Moreover, goal selections are uniform with slight biases towards specific suits on account of them having variable-length targets. For example, participants selected the hat suit significantly lower than the suits *cat* or *car* in the low disparity condition, perhaps accounted for by preferences for the options at extremes (in terms of target length). Fig 9C, 9D, and 9E show behavioral signatures from experiments 1 and 2 replicated in the new task.

Model comparisons for all models are shown in Fig 10. TD momentum provides the best account of participants' choices in the task using AIC, BIC, and cross-validation measures, as shown in Fig 10A. Fig 10B and 10C show behavioral signatures derived from model-generated datasets contrasted with participants' behavior. The suit selections in Fig 10B show an interesting distinction between TD-momentum and prospective agents. The prospective agent prefers the *cat* suit more than other suits across all conditions, as it has the smallest target of all suits and thus the lowest absolute distance to the target compared to the others. The hybrid agent, despite accounting for retrospective influences, shows the same

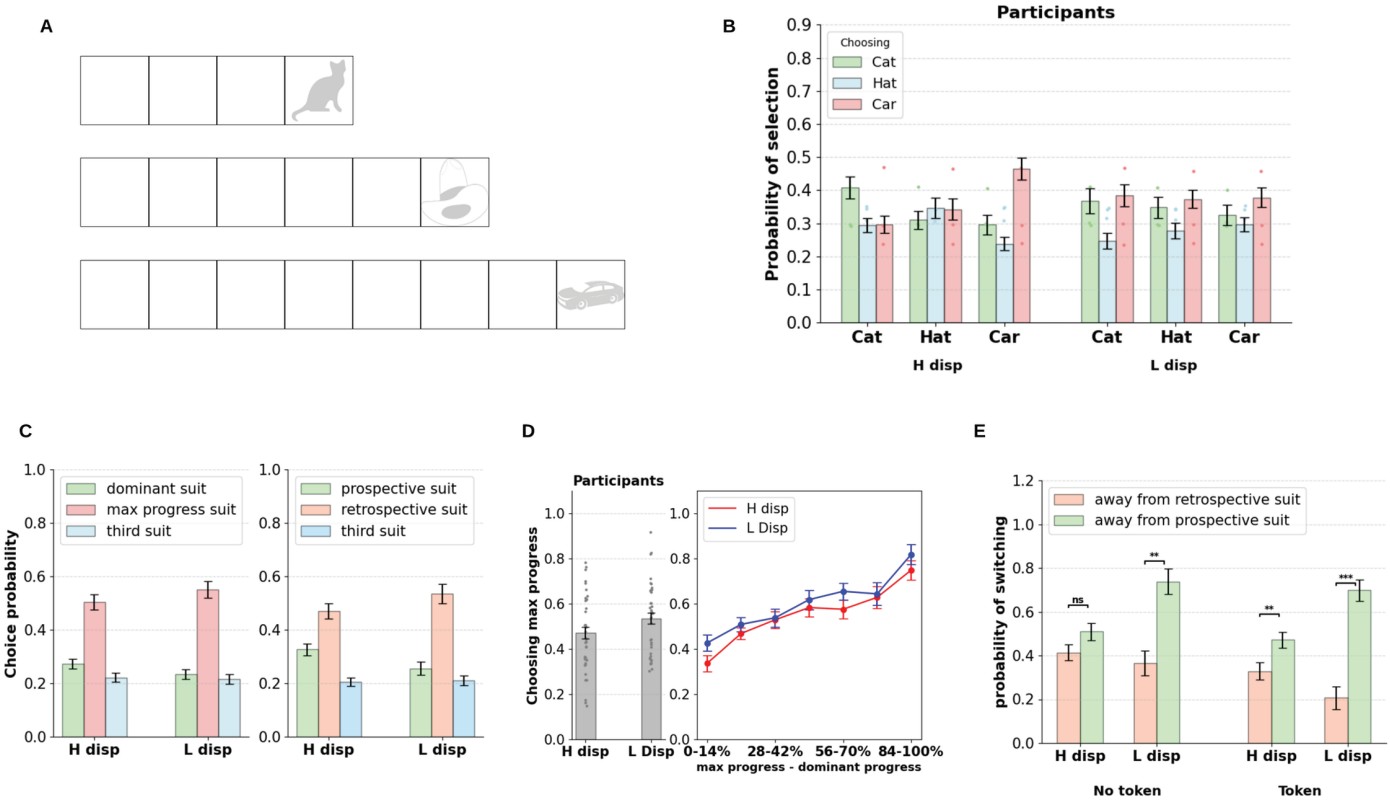

**Fig 9. Experiment 4: Variable targets for goals A.** Slot configuration for experiment 4. **B.** Suit selections divided by high disparity (H disp) and low disparity (L disp) conditions and sub divided by the dominant token type (cat/hat/car). **C, D, E.** Behavioral signatures (suit selections, retrospective choice, switching patterns) replicated from experiments 1 and 2. **Figure credits.** We used images from https://openclipart.org/ to generate this figure. See the Task Design section in Methods for full image credits.

pattern of preferences as that of the prospective agent (see S14 Fig). However, an optimal prospective agent shows no such preference for the smallest target suit as it perfectly incorporates token probabilities while estimating future prospects (see S15 Fig). The prospective agent fitted to the behavior of the participants shows the bias that indicates the inefficacy of the model in predicting the goal selections of the participants. Human participants and the TD-momentum model fitted to behavior do not show such a bias. Likewise, in Fig 10C, the TD-momentum model shows retrospective choice patterns that more closely resemble human behavior relative to the prospective agent. The TD-momentum model did not fully capture the specific bias against the *hat* suit displayed in the *L disp* condition; nevertheless, its distribution of goal selections appears to be closer to the pattern found in the choices of human participants. Finer deviations in goal selection patterns between the TD-momentum model and human behavior can potentially be attributed to idiosyncratic goal selection patterns in humans that are not factored into the model. The deviations could also arise from systematic biases (such as a preference for options at the extremes), which could be included in the model as inductive bias components.

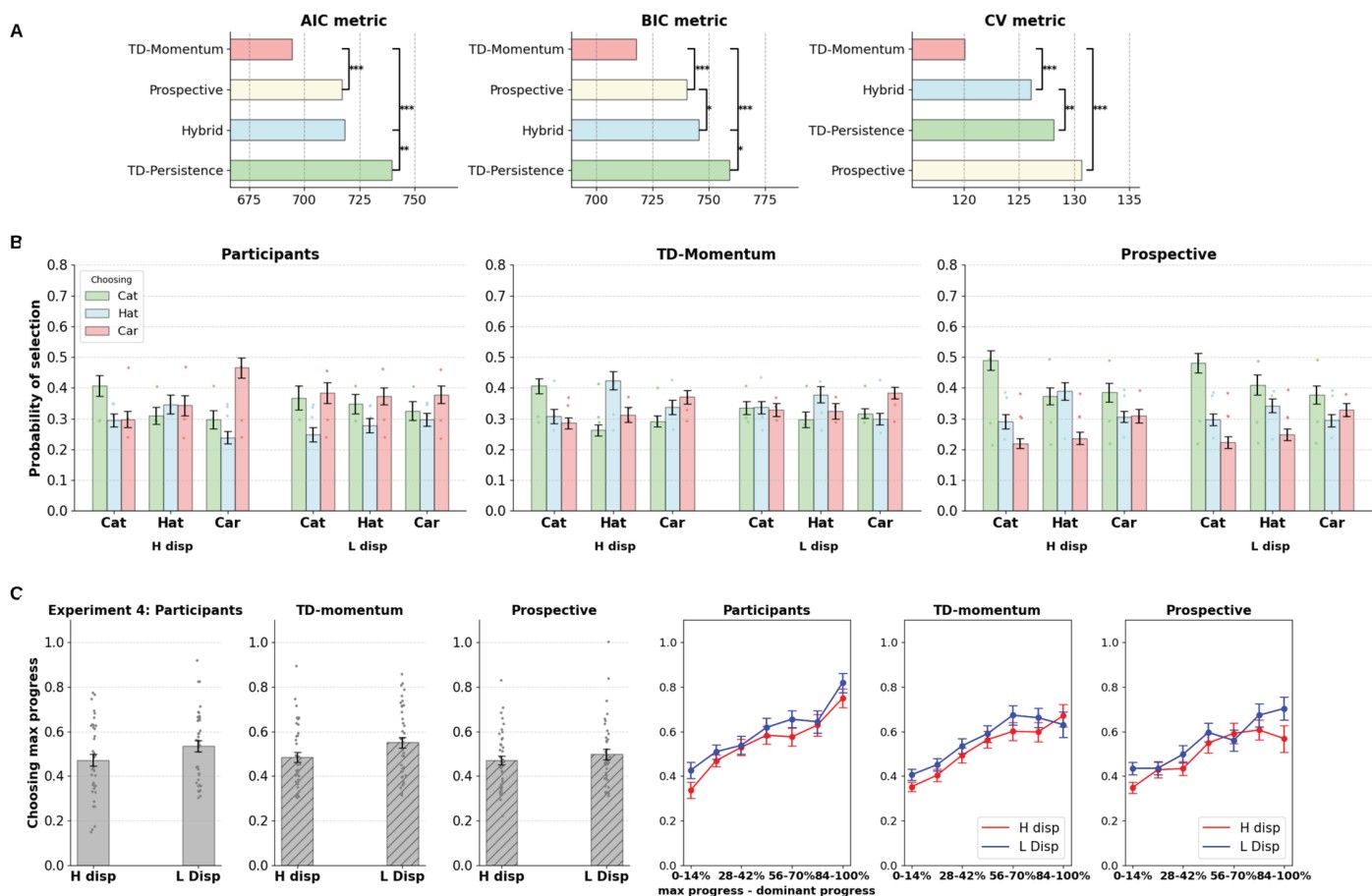

**Fig 10. Experiment 4: Model comparisons A.** TD-momentum outperforms other models in AIC, BIC, cross validation metrics. **B.** Prospective agent shows bias towards the cat suit, while TD-momentum closely captures distribution of goal selections. **C.** TD-momentum captures retrospective choice patterns better than the prospective agent.

### Algorithmic complexity and a normative account of momentum.

We provided evidence that incorporating the momentum of a goal as a proxy for its value can account for suboptimal switching and undue retrospective valuation in temporally-extended goal pursuit. However, is momentum-based decision making necessarily maladaptive?

Evidently, there are scenarios where reliance on momentum as a measure of goal value is suboptimal. When a goal with a considerable history of progress is abruptly decreased in value, its drop in prospective value is drastic, however, its momentum is sustained for a period beyond its decrease in value, whereupon reliance on momentum leads to suboptimal persistence. Conversely, when alternative goals are enhanced in value, their prospects are suppressed as their momentum is lower because of their lack of history.

If we revert to the interpretation of the term as it is used to describe the physical properties of objects, it is rare that we observe such drastic changes in momentum in a world where continuity is the norm. Consider the momentum of a car driving on a freeway, which is a continuous variable with occasional rises and dips; only in rare cases of collision do we notice an abrupt drop in momentum.

In a framework of continuity, where the recent past greatly predicts the near future, the momentum model offers a more efficient mode of computation and thereby an adaptive advantage. TD-momentum and the prospective agent both have identical parametric complexity (6 free parameters), yet their algorithmic complexities are different. Prospection relies on expensive model-based computations—keeping track of token-outcome expectations of each goal and recursively extrapolating them into the future. Momentum, on the other hand, maintains a model-free account of the past and uses it to extrapolate it to the future. The momentum model, without incorporating task contingencies, leads to suboptimalities, especially when there are jumps and discontinuities in prospects, as was the case in our task. The optimal performance attained by the TD momentum in the task is significantly lower than that of the prospective agent (see S7 Fig).

To test this account further, we simulated several variants of the task in which the prospects of the three goals undergo gradual shifts through Gaussian random walks. We estimated the optimal performance achieved in each of these variants by both modes of decision-making—momentum and prospection—and found that there is little difference in performance between the two algorithms (see Fig 11). However, when sudden reversals of the option probabilities are incorporated, the momentum model falls behind the prospective agent in performance.

Momentum computations can be efficient, yet in scenarios with sudden upheavals of fortunes, as is common in quotidian experiences, they can lead to suboptimal decisions. In addition, momentum computations and concomitant goal persistence can have adaptive features when there is uncertainty in the environment. Even when an option is devalued, one might

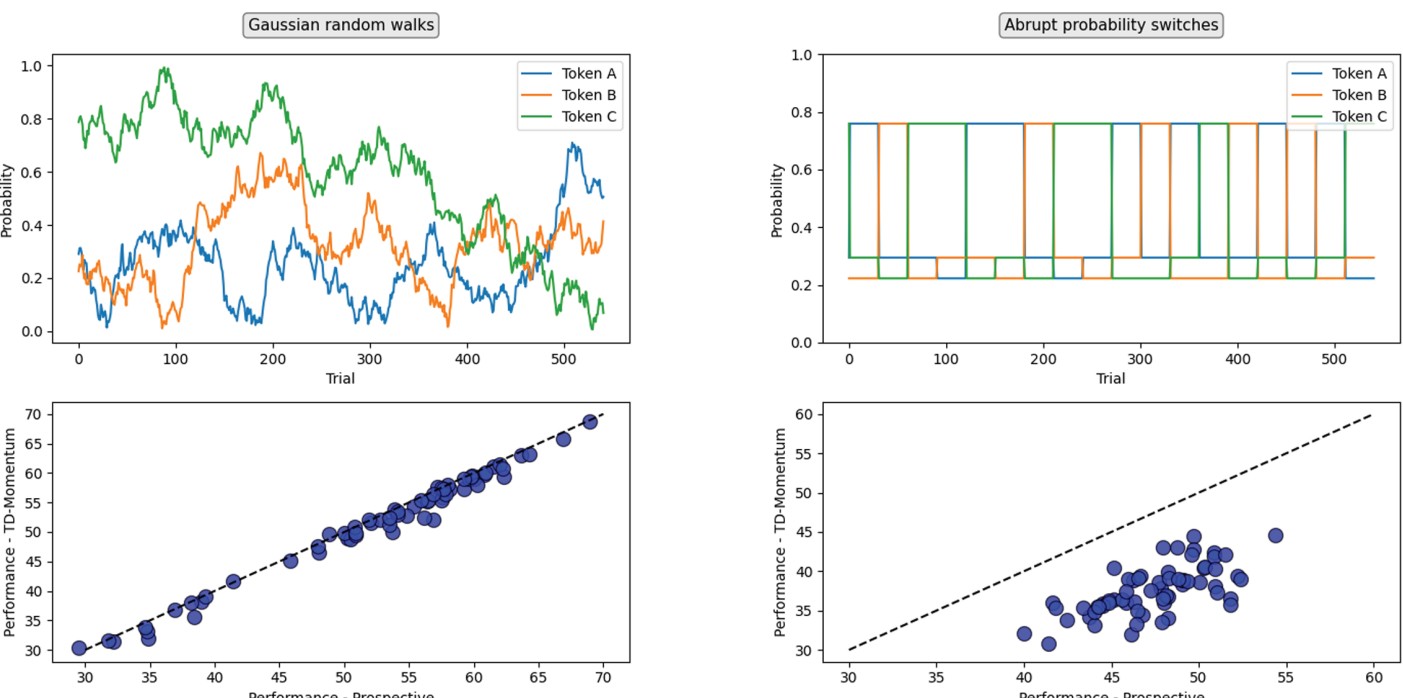

**Fig 11. Normative account of momentum in goal pursuit.** Optimal performance attained by the TD-momentum agent contrasted with that of the prospective agent when the token-outcome probabilities change in Gaussian random walks (left) and when they change in abrupt probability reversals (right). Each point in the plot refers to a variant of the task with randomly generated token contingencies; a sample task design was illustrated for both random walks and probability reversals.

be uncertain when it might be valued again, or whether viability of alternative options would remain stable enough to warrant a switch. Furthermore, persistence in goals can have other advantages in social settings, such as stable inference of one's goals by peers [2,34], all the while avoiding the inefficiency of incurring cognitive costs of goal switching.

## Discussion

Using a novel paradigm in which participants are free to pursue any of three temporally extended goals, we observed that participants exhibited a strong bias toward retrospectively persisting with goals with greater accumulated progress at the expense of goals with higher prospective benefits. The nature of this bias, which we call a retrospective bias—was highly heterogeneous in our pool of participants, with individuals manifesting varying degrees of retrospective influence on the selection of extended goals.

We manipulated block contingencies in the task so that there is a dominant goal that is most favorable to pursue in a given block that is not viable to pursue once the block shifted. This manipulation created situations in which participants made partial progress toward specific goals only to discover that they were no longer feasible shortly afterward. The optimal strategy would be to switch to an alternative option. However, we observed that participants demonstrated undue resistance in making such switches, which compromised performance on the task. Persistence towards a goal increased with an increase in accumulated progress. Participants also demonstrated sensitivity to prospective values, with the proportion of retrospective choice diminishing with an increase in the disparity of prospective values between options.

We conducted a variant of the task giving explicit instructions on the contextual prospects of different goals. By alerting the participants about the probability structure within a block, we expected that the inference of the identity of the dominant suit would be relatively easier and more salient. We noted that such explicit instructions served to diminish retrospective influences. However, they remained potent in driving goal selection. In this variant, participants continued to show suboptimal persistence toward the goals with greater accumulated progress despite having explicit knowledge about the prospects of alternative goals.

We explain this phenomenon by turning to the metaphor of momentum, aided by the simple fact that the momentum of a goal changes more slowly than its expected value. Momentum—a variable derived from physics defined as a product of mass and velocity—changes its magnitude with changes in velocity. Inertial mass acts as a moderating factor, tempering the impact of velocity on momentum. In this context, we draw a parallel by associating the velocity variable with the speed of progress toward the target and the mass variable with the prior progress accumulated in the task. When the accrued progress is substantial, any variation in the progress speed has a minor impact on the momentum's magnitude compared to when the progress is minimal. If participants prioritize the momentum of the goal over its expected value, this preference could explain their increased persistence towards the goals that have made significant progress towards the target. To give a computational account of this phenomenon, we introduced a novel TD-momentum algorithm that adjusts participants' momentum expectations for various goals using a temporal difference learning rule.

We compared the TD momentum algorithm with alternative accounts, including an inefficient prospective agent, a hybrid of prospective and retrospective agents, and a simple persistence-guided reinforcement learning of goal expectations. We hypothesized that momentum lends a better account of human persistence in goal valuation, as being driven by retrospective influences of past reinforcements accrued in goal pursuit, extending beyond

mere choice repetition or sunk-cost considerations. In fact, through our modeling, we demonstrate that incorporating momentum as a measure of goal value best accounts for behavior in the task better than its alternatives. This computational model outperforms alternative hypotheses in explaining the choice data and reproduces several behavioral patterns at both the population and individual levels.

The phenomenon of persistence in retrospective choice goes beyond task-switching costs or sunk costs, which are also potential drivers of goal persistence. Participants faced no explicit penalty for switching and can retain progress made in goals until the game's end and revisit them, hence the choice between stay or quit does not arise, except perhaps at the very end of the task. In addition, participants frequently revisited retrospective choices throughout the task, despite having switched away from them, indicating persistent goal valuation that goes beyond task switching costs. Nevertheless, to explain away persistence generated by task switching and/or sunk costs in a more formal way, we also included a choice kernel for goals in each of the computational models that we tested, whereby the tendency to stick with a particular goal is increased the more that goal is chosen in previous trials, acting as a proxy for a cumulative task switching effect. Our model fitting results show that momentum is critical to capture behavior above and beyond that explained by a cumulative task-switching cost effect. Thus, while our data do not rule out the contribution of one or another of these phenomena to goal selection and switching behavior in general, our findings show that such accounts are not sufficient to account for the bias toward retrospective choices we observed.

Our study goes beyond existing work on human goal commitment by providing a computational basis of overpersistence. [26] observed patterns of overpersistence during the pursuit of extended goals in a low-cost setting. However, their best-fitting model was not designed to account for the persistence biases displayed by the participants in the task. Instead, they quantified an individual persistence bias factor measured as a deviation from their prescribed goal selection model. Our model provides a plausible mechanistic account for participants' persistence patterns during goal selection. In support of this claim, the model parameters we fit to individual participants were found to correlate with the extent of individual retrospective preferences manifested in the task. Hence, our momentum model provides a computational account of human persistence in goal selection.

Although the momentum-based model accounts for cognitive biases manifested in the pursuit of temporally extended goals, it begs the question of why humans employ a suboptimal algorithm in pursuing extended goals. We also provide a normative explanation for the role of momentum in the pursuit of goals, indicating through simulations that such a strategy is not always maladaptive. In situations involving gradual shifts in the viability of the goal, interpolating future value through momentum is parsimonious, with little depreciation in expected rewards compared to an optimal strategy. However, when shifts in goal viability are more abrupt, as is our task, the momentum-based strategy of goal selection can lead to suboptimal persistence with goals of high retrospective value. Moreover, other adaptive effects of persistence in partially observed environments with uncertainty in goal prospects could also favor momentum computations.

The approach we have developed here can be applied to the broader question of how humans adaptively self-regulate, since the ability to disengage from goals that are unattainable in the current context is a hallmark of self-regulation [35,36]. An inability to disengage from goals is routinely observed in healthy individuals, with more pronounced effects seen in psychiatric conditions such as impulsivity [37,38], depression [39,40], attention deficit hyperactivity disorder [41], and generalized anxiety disorder [42,43]. The present findings have potential relevance for understanding the failures of behavioral persistence seen in psychiatric disorders.

There are several limitations of the present work. One is that we introduced a number of simplifying assumptions into our analysis which, when discarded, could open up a realm of other potential algorithmic accounts. For example, in formulating the prospective agent, we assumed that the agent optimizes for immediate rewards delivered upon suit completion. However, it is possible that humans could employ subroutines that are pre-planned according to the selection of hierarchical options [44], such as committing to a goal for a fixed set of rounds. We also observed significant heterogeneity in human goal selection with evidence of the prevalence of other strategies (such as the TD-persistence model) in some individuals. Furthermore, we found evidence that instructions can moderate the influence of momentum computations on goal selection behavior. Thus, the momentum model serves as a coarse descriptor of human goal selection behavior, but not necessarily as a fully comprehensive account for the goal selection tendencies of each individual.

In conclusion, the present work provides insight into the computational mechanisms underlying flexible goal pursuit. Characterizing how humans assign subjective value to goals over time and determining how that assignment is influenced by multiple factors, such as trial history, context, reinforcement rates, and the presence of alternative goals, is a crucial first step toward understanding how humans set and commit to goals in everyday decision-making contexts. Such insights can ultimately be used to implement novel behavioral interventions and nudges to improve the efficiency of goal-directed decision making within and between individuals.

## Methods

### Ethics statement

The research described here was reviewed by the Caltech Institutional Review Board and deemed exempt from ongoing in-depth review (IR18-0796) due to the fact that the data were collected anonymously through the online platform Prolific and because the research was considered of no more than minimal risk. All participants provided consent for their online participation by ticking a box on an online consent form, but did not sign the consent form, as the data collection is anonymized.

### Experimental protocol

**Task Design.** We have three different suits in the game: *cat, hat, car*. There are six cards, two for each suit. The mappings are as follows: the *car* suit is assigned to *mat* and *basket* cards; the *hat* is mapped to *wardrobe* and *table* cards; the *car* is mapped to *key* and *baggage* cards. We inform the participants about the above mapping at the beginning of the game. We chose the card symbols to be semantically related to the token types and we conveyed those semantic descriptions to the participant for their ease of retrieval (for instance, "places where you would like to keep a *hat*"). We further tested participants' knowledge of card-token mappings through a quiz before the game began. The participants continued with the game only if they answered all the questions correctly; if not, they were sent to the beginning of the instructions for revision.

In each round of the game, we presented the participants with three cards (one for each suit type). The exact card displayed for a particular token type is shuffled across the rounds. We also shuffled the locations of the cards on the screen at random every round to prevent any mappings between goals and action key presses.

Throughout the game, cards serve to stand in for the suits, and to collect tokens for a specific suit, participants need to flip the cards associated with the suit. Through this design, our

objective was to eliminate the stimulus-related value mapping, which would likely emerge if we were to directly display the suit stimuli (say images of *cat*, *hat*, *car*) and to elicit the construct of goals in participants. If, say, a participant persists in the goal of pursuing the *cat* suit, they would choose the relevant cards, the *basket* card when it is available and the *mat* card when it is not. By making the associated cards appear at random, we mitigated stimulus-elicited persistence effects in our design.

In experiment 1, we designed three block conditions: 80-20, 70-30, and 60-40. There are 18 blocks in the game, 6 of each type. Each block has 30 rounds, and the game lasts 540 rounds in total. The block conditions are shuffled throughout the game. However, we ensured that the dominant suit is flipped across adjacent blocks, so that no two adjacent blocks have the same token with the highest availability. As our primary focus lay in investigating flexible goal switching across contexts, we sought to induce conditions where switching is optimal.

In experiment 2, there are two block conditions: 75-25, 55-45. There are 12 blocks in the game (6 for each type), so 360 rounds in all.

For experiments 1 and 2, we instructed the participants that suit probabilities are constant within a block but shift across blocks. We included a quiz during the instructions to ensure that they understand block switches and differing suit probabilities. We also emphasized during the instructions that the probabilities are associated with the suit types and that picking either card related to the suit gives the token with the same probability. This is to prevent participants from tracking the value associated with specific card stimuli (whether a specific card gives the token more often) and to prompt them to place values on the goals (suit types) instead. In the experiment, block transitions are signaled with a screen that displays the text "This is the beginning of a new block" for 2 seconds. We also added a notation indicating the current block number out of the total number of blocks (say, block 13/18). We informed the participants that each block lasts 30 rounds and indicated the round number at the beginning of each round (say, round 20).

In experiment 3, a variant of experiment 1, we provided explicit instructions on the current token contingency faced by the participants. At the start of each block, we specified the type of block (80-20/70-30/60-40) without explicitly informing which token has the highest availability. During signaling the start of a new block, we also indicated that it is a specific type of block (say 80-20). We designed this as a nudge to encourage flexible goal adaptation across blocks.

In experiment 4, we assigned different targets to different suits. The cat suit can be finished by accruing 4 cat tokens, while the hat and car suits can be finished by collecting 6 and 8 tokens, respectively. Dominant suits are counterbalanced in all token types, and token probabilities are adjusted accordingly. We constructed two block conditions, high disparity and low disparity blocks (*H disp*, *L disp*) where the inferior token takes 15 and 12 rounds (on average) to finish. The dominant token takes 8 rounds (on average) to finish in all blocks.

We added a goal probe at the end of every third round, to elicit their target suits at different points throughout the game. To avoid a forced choice between three options, we also included an 'undecided' prompt.

Figs 1 and 9 in the manuscript were generated using images from https://openclipart.org.

1. https://openclipart.org/image/800px/1198 (cat)
2. https://openclipart.org/image/800px/159907 (hat),
3. https://openclipart.org/image/800px/321286 (car),
4. https://openclipart.org/image/800px/203250 (basket),
5. https://openclipart.org/image/800px/153547 (mat),
6. https://openclipart.org/image/800px/29276 (table),

**Table 2. Participant recruitment.**

|  | Participants | Median time (mins) | Age range (mean ± std) | Female/Male |
|---|---|---|---|---|
| Experiment 1 | 44 | 31 | 18-64 (28.6 ± 9.5) | 13/31 |
| Experiment 2 | 49 | 24 | 20-68 (29.2 ± 8.8) | 20/29 |
| Experiment 3 | 67 | 33 | 19-52 (28.1 ± 7.6) | 32/35 |
| Experiment 4 | 39 | 26 | 19-57 (31.1 ± 10.0) | 18/21 |

7. https://openclipart.org/image/800px/26415 (wardrobe),
8. https://openclipart.org/image/800px/320342 (baggage),
9. https://openclipart.org/image/800px/59329 (key).

**Participants.** Table 2 shows the details of the recruitment of the participants for all experiments. Participants were recruited from the Prolific on-line platform and compensated with a base pay of $9 per hour, with a performance-based bonus of up to $3. We instructed the participants that their performance bonus is directly proportional to the number of suits completed by the end of the game.

We did not exclude any participants from the study. Chronologically, we conducted experiment 3 as our first study, followed by experiments 2, 1, 4.

## Computational modeling

**Goal probes.** We elicited goal probes from the participant every third round of the task. We relied on goal probe estimates to account for model predictions of goal persistence that is different from action selection. To ensure reliability in goal probes, we measured goal-action congruence (whether the participant made an action that is congruent to the goal in the round following the goal probe prompt. S2A Fig shows the goal-action congruence density histograms plotted across all experiments. The mean congruence in the target probes is 80%, 78%, 76%, and 73% for experiments 1 to 4.

We also estimated the proportion of the total number of switches that the participant made in the task that followed a goal probe. This is to verify whether the goal probe prompt itself influenced subsequent goal selection (either prompting a switch/prompting persistence in the specified probe). S2B Fig shows the mean proportion of switches are 31%, 32%, 29%, 32% for experiments 1 to 4. As the goal is probed every third round (in 33% of the rounds), the proportion of switches around the goal probes is in line with the chance expectation of 33%.

The congruence of goal action is high but not perfect in the participants. We anticipated this mismatch by our hierarchical assumption in our modeling, by dissociating participant goal and action selection. Participants can value a goal above other alternatives, but can temporarily explore other options before revisiting it. We found such patterns of revisiting in participants (Fig 3), indicating persistence in goal valuation. However, we still expect noise in goal probes in a portion of less-engaged participants (perhaps those showing low goal-action congruence) and that is a fundamental limitation in probing intrinsic goal states in participants. This noise could explain why the models do not capture the goal selection patterns in the task perfectly.

**Prospective and retrospective models. Prospective.** The prospective model updates the belief states of the current suit probabilities and rolls out the expected outcomes in the future.

The estimate of the probability of token collection for a suit $g$ can be computed using a delta learning rule.

$$M_g^{(t)} = M_g^{(t-1)} + \eta(I_g^{(t)} - M_g^{(t-1)}) \tag{13}$$

Using the probability of token outcome, the prospective agent rolls out expectations one step into the future, for two scenarios: receiving a token and moving one token closer towards the target, and not receiving the token and staying in the same state. The value estimate for a goal $g$ in a specific configuration state of the slots $s_g$ is calculated as follows:

$$Q_g(s_g^{(t)}) = M_g \gamma Q_g(s_g^{(t)} + 1) + (1 - M_g) \gamma Q_g(s_g^{(t)}) \tag{14}$$

where $s_g$ is the current state of the goal (number of tokens collected) and $\gamma$ is the temporal discounting factor that ensures sooner rewards are valued higher over later rewards. This algorithm prefers the suit that can be completed fastest in a given context.

**Retrospective.**

The retrospective model values the goals with the greatest progress.

$$Q_g(s_g^{(t)}) = \gamma Q_g(s_g^{(t)} + 1) \tag{15}$$

As $s_g$ is closer to the target, there is less discounting of rewards and therefore higher value. This formulation is chosen to ensure similar scaling and parameterization for prospective and retrospective models.

**TD-momentum.** Momentum in the context of goal valuation is defined as the product of the progress made and the speed of progress. Here, the learnable parameter is the speed of progress. The goal value is formulated as a function approximator for momentum as follows.

$$Q_g(s_g^{(t)} = m^{(t)}) = v_g^{(t)} m^{(t)} + b_g^{(t)} \tag{16}$$

where $Q_g$ is the value of a given suit $g$ in a specific slot configuration $s_g^{(t)}$, $m^{(t)}$ is the progress (fraction of slots filled) at a given instant $t$, $v_g$ is the parameter for the speed of progress and $b_g$ is a bias for the goal.

Depending on whether a token is received in a given round, the state of the goal is updated accordingly. The speed of progress is learned with the following temporal difference learning rule.

$$\delta = \gamma Q_g(s_g^{(t+1)}) - Q_g(s_g^{(t)}) \tag{17}$$

$$v_g^{(t+1)} = v_g^{(t)} + \alpha\delta \frac{dQ_g^{(t)}}{dv_g^{(t)}} = v_g^{(t)} + \alpha\delta m^{(t)} \tag{18}$$

$$b_g^{(t+1)} = b_g^{(t)} + \alpha\delta \tag{19}$$

We used a fixed initialization for $v_g = 0.5$ and $b_g = 0.1$ for all goals. We tested different initializations to assess their impact on the model fitting and found that they are not significantly impacted by the choice of function approximator initialization.

We ensured that all the parameters of the model are recoverable through parameter recovery analysis (see S4 Fig).

**Alternative hypotheses. TD Persistence.**

Our task can be solved by reducing it to a three-armed bandit task and finding the optimal bandit in each block that gives you the most rewards and persists with it until the end of the block. We assumed that this might be a plausible strategy in the experimental variant where we clearly inform the participants of the token contingencies of the block.

The token-outcome expectations from each goal, calculated using a delta rule, drive the choices in this algorithm. Once the value of a certain token sufficiently exceeds its alternatives, the persistence parameters in the action policy (described later) ensure persistence towards the token.

$$M_g^{(t)} = M_g^{(t-1)} + \eta \left( I_g^{(t)} - M_g^{(t-1)} \right) \tag{20}$$

$$Q_g \left( s_g^{(t)} \right) = M_g^{(t)} \tag{21}$$

**Hybrid.**

A potential mechanism to incorporate both prospective and retrospective influences is to assume a fixed weight arbitration model that weights contributions from both valuations and their combined influence drives decision making.

$$Q_g \left( s_g^{(t)} \right) = w * Q_g^{pros} + (1 - w) * Q_g^{retro} \tag{22}$$

**Common mechanism of goal and action selection.** All competing hypotheses of goal valuation followed a common action selection policy elucidated as follows.

**Algorithm 1 Algorithm for goal selection.**

```
1: Q_g: goal value where g ∈ {cat, hat, car}
2: β_g, β_a: softmax temperatures for goal and action selection
3: c_g: goal preservation/choice kernel
4: c: cost of switching between goals
5: G: current goal (none at start)
6: for block in (1, num_blocks) do
7:    Initialize card probabilities
8:    for round in (1, num_rounds) do
9:       Get goal values (Q_cat, Q_hat, Q_car)   // Using TD-momentum/prospective
10:      Calculate advantage of current goal G:
            A_G = Q_G − max_{g≠G} Q_g
11:      Choose to stay with/switch from the current goal:
            p_{stay;G} = 1/(1+exp(−c−c_G−β_g A_G))            // Choose goal
12:      In case of a switch, chose the new goal according to goal
   values:
            softmax_{β_g} {Q_{alt_1}, Q_{alt_2}}
13:      Update current goal G
14:      Update choice kernel of goal selection
15:      Report the goal G every third round
16:      Choose to act according to the current goal or explore
            p_{stay;G,A} = 1/(1+exp(−c−c_G−β_a A_G))           // Choose action
17:      In case of a switch, explore a new action according to goal
   values:
            softmax_{β_a} {Q_{alt_1}, Q_{alt_2}}
18:      Observe outcome
19:      Update goal-outcome probabilities                    // For
   prospective/hybrid/TD-persistence
20:      Update goal values
21:   end for
22: end for
```

Once the values of individual goals are computed, the advantage of the current goal, $A_G$, over its alternatives is

$$A_G = Q_G - \max_{c \neq G} Q_c \tag{23}$$

The agent chooses to stay on the current goal (target suit) with a probability $p_{\text{stay};G}$,

$$p_{\text{stay};G} = \frac{1}{1 + \exp(-c - c_G - \beta_g A_G)} \tag{24}$$

where $c$ is the switching cost, $\beta_g$ is the softmax temperature to scale the advantage. $c_G$ is the choice kernel where commitment towards the goal increases the more it is chosen in the past as updated through the delta-rule,

$$c_G = (1 - \alpha_c)c_G + \alpha_c c_G I_G \tag{25}$$

where $I_G$ is the indicator that the goal is chosen in the round.

When the agent decides to switch away from the current goal with probability $(1 - p_{stay})$, it selects a new goal from the two alternate goals depending on the values of the two alternate goals ($softmax(Q_{alt_1}, Q_{alt_2})$)

Once the goal is chosen, the participant chooses to perform an action in accordance with the goal (choosing the cards) with a probability, $p_{\text{stay};G,A}$, which is modulated by a separate softmax term $\beta_a$.

We chose separate softmax terms for actions and goals to allow goals to change at a different rate than actions. Participants can switch actions to explore other offers while staying on the same high level goal, and can choose to switch goals when the value of an alternative goal exceeds that of the current one. This framing can account for both the choices and the goal probes reported by the participants. Algorithm 1 shows the pseudocode for the common goal and action selection mechanism.

**Comparisons with simpler versions: Rescorla-Wagner + choice kernel.** We contrasted the models with a much simpler Rescorla-Wagner model embedded with a choice kernel.

$$Q_g^{(t)} = Q_g^{(t-1)} + \eta(r_g^{(t)} - Q_g^{(t-1)}) \tag{26}$$

$$c_g(t) = c_g(t-1) + \alpha_c(I_g(t) - c_g(t-1)) \tag{27}$$

where $Q_g(t)$ is the expected value of token rewards for each goal, $r_g(t)$ is the binary reward of receiving a token, $c_g(t)$ is the choice kernel for the goal, $I_g(t)$ is the binary indicator value for the goal chosen each round.

The goal selection probability is given as

$$p_g(t) = \frac{\exp(\beta Q_g(t) + \beta_c c_g(t))}{\Sigma_{k \in G} \exp(\beta Q_k(t) + \beta_c c_k(t))} \tag{28}$$

**Model fitting procedure.** We fit both participant card selections and goal probes in all models except for the Rescorla-Wagner model, where we just fit participants' choices (where there is no notion of an extended goal).

The likelihood of a goal probe is estimated only in the rounds where goal probes are presented in the task (i.e., every third round), whereas the likelihood of an action is estimated in every round. In rounds where the participant does not report their goal, the goal remains

latent, and the model estimates selection probabilities across all possible goals. Goal and action likelihoods are then computed by integrating over these latent goal probabilities. The full likelihood estimation process for goal and action selections in the task is summarized as follows.

1. Estimating goal selection likelihood:

i. If the participant's goal is probed in round $k$ (say the goal reported is *cat*)

$$\Pr(Goal_k = cat) = 1; \Pr(Goal_k = hat) = 0; \Pr(Goal_k = car) = 0; \tag{29}$$

ii. For round $k + 1$, the latent goal probabilities are estimated given latent goal in $k$:

$$\Pr(Goal_{k+1} = G) = \Pr(Goal_{k+1} = G | Goal_k = cat) \tag{30}$$

Conditional probability of choosing a goal $g$ in round $t$ given the previous goal $G$ in round $t-1$ is defined as:

$$Pr(Goal_t = g | Goal_{t-1} = G) = \begin{cases} p_{stay}, & g = G \quad \text{(stay)} \\ (1 - p_{stay}) * softmax(Q_{alt_1}, Q_{alt_2})[g], & g \neq G \quad \text{(switch)} \end{cases}$$

where $p_{stay}$ is estimated as described in Equation (20), and the softmax is computed over the two alternative goal values.

iii. For round $k + 2$, the latent goal probabilities are estimated given latent goal probabilities in $k + 1$:

$$\Pr(Goal_{k+2} = G) = \sum_g \Pr(Goal_{k+2} = G | Goal_{k+1} = g) * \Pr(Goal_{k+1} = g) \tag{31}$$

iv. The goal is then probed again in round $k + 3$, with the goal selection probabilities computed given the latent goal probabilities in $k + 2$. This estimation is used to compute the likelihood of goal selection for the participant's goal probe in round $k + 3$.

$$\Pr(Goal_{k+3} = G) = \sum_g \Pr(Goal_{k+3} = G | Goal_{k+2} = g) * \Pr(Goal_{k+2} = g) \tag{32}$$

v. The goal probabilities in round $k + 3$ are aligned with the latest goal probe (as in step (a)).

2. Estimating action selection likelihood:

$$\Pr(Action_t = A) = \sum_g \Pr(Action_t = A | Goal_t = g) * \Pr(Goal_t = g) \tag{33}$$

One limitation of this procedure is that we hold the goal probe reported by the participant as the true value of the latent goal variable. However, the goal reports of participants could be noisy in disengaged participants. The participant goal probe statistics reported in S2A and S2B Fig can alleviate some of these concerns given the high congruence of goal action seen in the participant data.

TD-persistence has 5 free parameters, TD-momentum and prospective agent have 6, and hybrid agent has 7. For each individual participant, we estimate the parameters for each of

the competing models to maximize the likelihood of their action choices and goal probes by employing a tree-structured parzen estimator (TPE) sampler, a Bayesian optimization algorithm implemented in optuna [45], an open source hyperparameter optimization framework in Python. We estimated the parameters that maximized the likelihood over 100 random starts.

We followed the same procedure for fitting behavior and for recovering parameters and models.

**Parameter recovery.** For parameter recovery analysis, we sampled 1000 parameter vectors from a range of parameters ($\alpha \in [0,1], c_g \in [-1,1], \beta_g \in [0,10], \beta_a \in [0,10], \gamma \in [0.6,1.0], \alpha_c \in [0,1]$). We generate synthetic behavior for each parameter vector using the TD-momentum model and recover the parameters by fitting the model to the simulated data. We optimized both the likelihoods of selection of the goals and the actions while fitting the data, where the simulated agent reported the goal every third round as the participants did. We plotted the fitted parameters and the generated parameters and reported correlations (see S4 Fig). Most of the parameters are recoverable with high correlation coefficients. However, low discount factors closer to 0.5 do not recover well; however, the discount factors of the participant fits are mostly in the range $[0.9,1.0]$, so we considered the range $[0.6,1.0]$ for parameter recovery.

**Model recovery.** We performed a model recovery analysis on the competing models. We sampled 100 random parameter vectors from parameter ranges derived from individual participant fits. For each of the four algorithms (TD-momentum, TD-persistence, prospective, hybrid), we generated choice data using the sampled parameters. We then fit the generated data to each of the four models and identify the winning model through BIC score comparisons. We plotted the confusion matrices where the rows correspond to the simulated model and the columns correspond to the recovered model (see S5 Fig). The left confusion matrix indicates the probability of fitted model given the simulated model, whereas the right matrix indicates the probability of simulated model given the fitted model. In all cases, we observed that the data generating model provides the best explanation for the choice data. However, there appears to be some confusion between the prospective and hybrid models.

**Normative simulations.** For the normative account, we assumed three starting positions for the availability of three options: $\sim \mathcal{U}(0.2,0.3), \sim \mathcal{U}(0.2,0.3), \sim \mathcal{U}(0.7,0.8)$. We simulated two variants: token contingencies that gradually changed through Gaussian random walks ($\sim \mathcal{N}(0,0.025)$) and abrupt switching of contingencies with probabilities shuffling every 30 rounds.

We computed the optimal performance achieved by each algorithm with parameters estimated by Bayesian optimization.

## Supporting information

**S1 Appendix. Experimenting with variants of the prospective model.**
(PDF)

**S1 Fig. Goal selections with no conflict. A.** When the dominant and maximum progress suits are the same, participants predominantly choose the dominant/maximum progress option because it is devoid of goal conflict between accrued progress and the current rate of progress. **B.** Similar pattern of preference is shown when suits are classified as prospective and retrospective (based on predictions of the prospective agent optimized for the task).
(TIF)

**S2 Fig. A. Goal-action congruence.** Figure shows the frequency distribution of the proportion of rounds where the participant performed action in congruence with the objective probe in the next round. **A. Switches after goal probes** Frequency distribution of the proportion of the total number of switches made by participants that occur in the round following the goal probe.
(TIF)

**S3 Fig. A. Proportion of dominant choice traced with task progress.** Blocks are split into groups of 3 to track progress throughout the task. There is no indication of increased persistence at the end of the task (last three blocks) compared to the beginning (first three blocks), except in experiment 2 where there is a marginal decrease in preference towards the dominant suit at the end. **B.** Proportion of the choice of dominant suit with no grouping of blocks. Oscillations in dominant choice proportion correspond to the block condition encountered.
(TIF)

**S4 Fig. TD Momentum parameters are recoverable.** 1000 randomly generated parameter vectors were used to generate synthetic behavior and were recovered by fitting the model (recovery analysis in experiment 1).
(TIF)

**S5 Fig. Competing computational hypotheses are individually recoverable.** Data generating model offers the best explanation for all competing hypotheses (recovery analysis in experiment 1).
(TIF)

**S6 Fig. The simple variant of prospection with symmetric learning rates outperforms other variants (model fits for experiment 1)**.
(TIF)

**S7 Fig. Optimal performance of models.** Number of suits collected by each agent broken down by block type when their parameters are optimized for task performance. Prospective and TD-persistence agents perform the best, and the retrospective agent gives the lowest performance. TD momentum performs better than the retrospective agent but is short of the other agents.
(TIF)

**S8 Fig. Heterogeneity in strategies in the version with explicit instructions.** 40% of participants in the version with explicit instructions about the current block type were better accounted for by TD-persistence, with the rest 60% favoring TD-momentum according to BIC criteria. Lower retrospective bias and higher task performance is observed in the group that is better accounted by TD-persistence. Retrospective bias of the participants explained by TD-persistence follows the similar pattern as demonstrated by simulations of the algorithm.
(TIF)

**S9 Fig. Learning effects within blocks.** Figure shows the participants' dominant suit choice probability in the first and second halves of the block across all experiments replicated by TD-momentum.
(TIF)

**S10 Fig. Model predictions of the prospective agent**. Prospective agent captures general trends in preferences towards maximum progress suit and switching patterns in the task.
(TIF)

**S11 Fig. Model predictions of the hybrid agent**. Hybrid agent captures general trends in preferences towards maximum progress suit and switching patterns in the task.
(TIF)

**S12 Fig. Model predictions of the TD-persistence agent**. TD-persistence agent does not capture preferences towards maximum progress and also shows increased preference towards switching away from the retrospective agent in contrast to the participants.
(TIF)

**S13 Fig. Model predictions of the Rescorla-Wagner agent**. Rescorla-Wagner agent neither captures preferences towards maximum progress nor patterns of switching seen in participants.
(TIF)

**S14 Fig. Experiment 4: Goal selection patterns in hybrid and prospective agents.** Both hybrid and prospective agents show a preference for the cat suit (smallest target) across conditions.
(TIF)

**S15 Fig. Experiment 4: Goal selection patterns comparing optimal prospective agent and behavior-fitted prospective agents.** Optimal prospective agent prefers optimal suit in each condition. Prospective agent fitted to participant data displays bias towards the cat suit.
(TIF)

## Acknowledgments

We thank Antonio Rangel, Colin Camerer, Caltech SDN graduate program students, and O'Doherty lab members for their valuable feedback and support throughout the development of the manuscript.

## Author contributions

**Conceptualization:** Sneha Aenugu, John P. O'Doherty.

**Data curation:** Sneha Aenugu.

**Formal analysis:** Sneha Aenugu.

**Investigation:** Sneha Aenugu.

**Methodology:** Sneha Aenugu, John P. O'Doherty.

**Project administration:** Sneha Aenugu, John P. O'Doherty.

**Resources:** John P. O'Doherty.

**Software:** Sneha Aenugu.

**Validation:** Sneha Aenugu, John P. O'Doherty.

**Visualization:** Sneha Aenugu.

**Writing – original draft:** Sneha Aenugu.

**Writing – review & editing:** Sneha Aenugu, John P. O'Doherty.

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
