## [Decision Letter · Decision Letter 0]

28 Sep 2024

Dear Ms Aenugu,

Thank you very much for submitting your manuscript "Building momentum: A computational model of persistence in long-term goals" for consideration at PLOS Computational Biology.

As with all papers reviewed by the journal, your manuscript was reviewed by members of the editorial board and by several independent reviewers. In light of the reviews (below this email), we would like to invite the resubmission of a significantly-revised version that takes into account the reviewers' comments.

We cannot make any decision about publication until we have seen the revised manuscript and your response to the reviewers' comments. Your revised manuscript is also likely to be sent to reviewers for further evaluation.

Sincerely,

Peter E. Latham

Academic Editor

PLOS Computational Biology

Daniele Marinazzo

Section Editor

PLOS Computational Biology

Reviewer's Responses to Questions

**Comments to the Authors:**

Reviewer #1: See attachment.

Reviewer #2: Please see the attachment.

**Have the authors made all data and (if applicable) computational code underlying the findings in their manuscript fully available?**

Reviewer #1: Yes

Reviewer #2: None

PLOS authors have the option to publish the peer review history of their article (what does this mean?). If published, this will include your full peer review and any attached files.

Reviewer #1: No

Reviewer #2: No
---

## [Decision Letter · Decision Letter 1]

13 Feb 2025

PCOMPBIOL-D-24-01340R1

Building momentum: A computational model of persistence in long-term goals

PLOS Computational Biology

Dear Dr. Aenugu,

I called this a major revision, but it seems that i's more about writing than the science. So hopefully it will be straightforward. Hopefully. ;)

Peter

--formal letter follows

Thank you for submitting your manuscript to PLOS Computational Biology. After careful consideration, we feel that it has merit but does not fully meet PLOS Computational Biology's publication criteria as it currently stands. Therefore, we invite you to submit a revised version of the manuscript that addresses the points raised during the review process.

Please submit your revised manuscript within 60 days Apr 15 2025 11:59PM. If you will need more time than this to complete your revisions, please reply to this message or contact the journal office at ploscompbiol@plos.org. Please include the following items when submitting your revised manuscript:

We look forward to receiving your revised manuscript.

Kind regards,

Peter E. Latham

Academic Editor

PLOS Computational Biology

Daniele Marinazzo

Section Editor

PLOS Computational Biology

**Journal Requirements:**

1) Regarding Figures 1 and 9 :Thank you for indicating that "We redid the images using open-source resources that you recommended (https://openclipart.org/)." Please add this information in the figures legends.

**Reviewers' comments:**

Reviewer's Responses to Questions

Reviewer #1: I would like to compliment and thank the authors for taking great care in addressing most of the questions from both us reviewers.

However, I feel that the task contingencies are still not explained clearly, which has been a source of a major misunderstanding on my side -- about which I had an exchange with the editor.

My original question --probably unclearly formulated-- was about whether or not the *card* chosen by the participant had any effect on the performance.

What I meant by *card*, confusingly, was the *token represented by the card* (car, hat, cat), and not the specific picture on the card (mat, wardrobe, key, etc).

For instance, in Fig 1, in the first block of Exp 1 one has

p(car) = 0.8

p(cat) = 0.2

p(hat) = 0.2

According to my understanding, this meant that whichever card (the actual card, with the token-related symbol on it) the participant picked, the probability of obtaining a token (say, "car") would always be the same (0.8 for the "car").

Instead, as I am aware the editor confirmed with you, these probabilities should be expressed more rigorously as *conditional probabilities* as follows:

p(car token|car related card) = 0.8

p(car token|cat related card) = 0

p(car token|mat related card) = 0

p(cat token|car related card) = 0

p(cat token|cat related card) = 0.2

p(cat token|mat related card) = 0

p(mat token|car related card) = 0

p(mat token|cat related card) = 0

p(mat token|mat related card) = 0.2

In the new text, we read:

"Participants receive tokens upon flipping cards with a given probability. If they

receive the token for a suit type in a round, it goes into its designated slots and the

participant proceeds to play another round."

A whole paragraph later (!), we have a description of the probabilities:

"Furthermore, we designed blocks in the game so that there is a dominant suit in

each block (counterbalanced across all suit types), such that flipping its cards gives

tokens with the highest probability (in Figure 1D, the cat suit is the dominant one

with 0.8 probability in block N ). The other two suits have identically inferior token

probabilities (both hat and car with 0.2 probability)."

It currently takes 5 (!) paragraphs to get to the to these probabilities.

They should come immediately (right after "Participants receive tokens upon flipping cards with a given probability") with the help of equations, and also with a clear graphical representation of the *conditional* probabilities in Fig. 1.

This points back to one of the first comment I made in the first round of revisions: the exposition should be more succinct and rely more on mathematical formulas.

I expect the readers of PLOS Computational Biology to be generally well versed in math, and this should not be an obstacle; rather, I feel it would help most readers go through the setup and the results.

Currently, I feel that the exposition is too verbose, at the cost of readability.

For this reason, I would strongly recommend to make further edits in this direction, to make justice to an otherwise interesting and solid work.

Reviewer #2: I am pleased that the authors addressed most of the reviewers’ points in their new submission, which I find much improved compared to the original one. The current manuscript provides a clearer description of the methodologies, more adequately integrates existing research, and controls for alternative explanations thanks to newly added analyses. However, a couple of points remain to be addressed.

One point that remains unclear is regarding the fitted models’ likelihood with respect to participants’ goals. In particular, the authors affirm that in rounds where participants do not explicitly declare goals the model selects goals by itself based on their relative values. Such goals are then used to estimate the likelihood of participants’ chosen actions, which are known. However, the participants’ goals are latent variables, meaning standard model fitting methods based on likelihood may be inappropriate unless integrating over the latent variables across all trials (see e.g., Rmus et al., 2024 Plos Comp Bio). The authors should thus resort to alternative, likelihood-free model fitting methods, or explain why their approach works in the context of their task.

Moreover, is it correct that the model (algorithm 1) decides whether to switch goals based on the values of the current and the best alternative goal, but that in the case of a switch, the model could select the third available goal? If so, could the authors please justify this choice?

I also have some minor suggestions.

In some figures with model simulations, e.g., 2 and 4, the authors add the “optimal” specification to prospective and retrospective models to specify these models were not fit on participants’ behavior. However, I believe optimized would be a more appropriate term, especially for the retrospective model, since it is not optimal for task performance.

In addition, could the authors please state somewhere how many iterations per real participant were run by the models in order to produce simulated data?

On page 14, the authors state the prospective agent shows a “clear” interaction of performance with block type. However, I think “significant” would be a more appropriate adjective.

In general, I think the authors would benefit from additional proofreading and editing, as I noticed several typos/inconsistencies that should be corrected before the final submission.

**Have the authors made all data and (if applicable) computational code underlying the findings in their manuscript fully available?**

Reviewer #1: Yes

Reviewer #2: None

PLOS authors have the option to publish the peer review history of their article (what does this mean?). If published, this will include your full peer review and any attached files.

Reviewer #1: No

Reviewer #2: No

**Figure resubmission:**
---

## [Decision Letter · Decision Letter 2]

14 Apr 2025

Dear Ms Aenugu,

We are pleased to inform you that your manuscript 'Building momentum: A computational model of persistence toward long-term goals' has been provisionally accepted for publication in PLOS Computational Biology.

Best regards,

Peter E. Latham

Academic Editor

PLOS Computational Biology

Daniele Marinazzo

Section Editor

PLOS Computational Biology

Reviewer's Responses to Questions

**Comments to the Authors:**

Reviewer #1: I am pleased to see that the authors have addressed all reviewers' comments and questions. The task is much more clearly explained now. I would recommend publication in Plos Comp Bio.

Reviewer #2: I am satisfied with how the authors have addressed the points I raised and have no further comments.

**Have the authors made all data and (if applicable) computational code underlying the findings in their manuscript fully available?**

Reviewer #1: Yes

Reviewer #2: None

PLOS authors have the option to publish the peer review history of their article (what does this mean?). If published, this will include your full peer review and any attached files.

Reviewer #1: No

Reviewer #2: No